# Generalization Bounds for Meta-Learning via PAC-Bayes and Uniform Stability

**Alec Farid**      **Anirudha Majumdar**
Department of Mechanical and Aerospace Engineering, Princeton University
{afarid, ani.majumdar}@princeton.edu

## Abstract

We are motivated by the problem of providing strong generalization guarantees in the context of meta-learning. Existing generalization bounds are either challenging to evaluate or provide vacuous guarantees in even relatively simple settings. We derive a probably approximately correct (PAC) bound for gradient-based meta-learning using two different generalization frameworks in order to deal with the qualitatively different challenges of generalization at the "base" and "meta" levels. We employ bounds for uniformly stable algorithms at the base level and bounds from the PAC-Bayes framework at the meta level. The result of this approach is a novel PAC bound that is tighter when the base learner adapts quickly, which is precisely the goal of meta-learning. We show that our bound provides a tighter guarantee than other bounds on a toy non-convex problem on the unit sphere and a text-based classification example. We also present a practical regularization scheme motivated by the bound in settings where the bound is loose and demonstrate improved performance over baseline techniques.

## 1   Introduction

A major challenge with current machine learning systems is the need to acquire large amounts of training data in order to learn a new task. Over the past few decades, meta-learning [62, 70] has emerged as a promising avenue for addressing this challenge. Meta-learning relies on the intuition that a new task often bears significant similarity to previous tasks; hence, a learner can learn to perform a new task very quickly by exploiting data from previously-encountered related tasks. The meta-learning problem formulation thus assumes access to datasets from a variety of tasks during meta-training. The goal of the meta learner is then to learn inductive biases from these tasks in order to train a base learner to achieve few-shot generalization on a new task.

Over the past few years, there has been tremendous progress in practical algorithms for meta-learning (see, e.g., [61, 55, 25, 32]). Techniques such as model-agnostic meta-learning (MAML) [25] have demonstrated the ability to perform few-shot learning in a variety of supervised learning and reinforcement learning domains. However, our theoretical understanding of these techniques lags significantly behind successes on the empirical front. In particular, the problem of deriving *generalization bounds* for meta-learning techniques remains an outstanding challenge. Current methods for obtaining generalization guarantees for meta-learning [5, 34, 77] either (i) produce bounds that are extremely challenging to compute or (ii) produce vacuous or near-vacuous bounds in even highly simplified settings (see Section 5 for numerical examples). Indeed, we note that existing work on generalization theory for meta-learning techniques do not explicitly report numerical values for generalization bounds. This is in contrast to the state of generalization theory in the supervised learning setting, where recent techniques demonstrate the ability to obtain non-vacuous generalization guarantees on benchmark problems (e.g. visual classification problems [24, 78, 54]).

The generalization challenge in meta-learning is similar to, but distinct from, the supervised learning case. In particular, any generalization bound for meta-learning must account for *two levels* of

35th Conference on Neural Information Processing Systems (NeurIPS 2021).

generalization. First, one must account for generalization at the base level, i.e., the ability of the base learner to perform well on new data from a given task. This is particularly important in the few-shot learning setting. Second, one must account for generalization at the meta level, i.e., the ability of the meta learner to generalize to new tasks not encountered during meta-training. Moreover, the generalization performance at the two levels is coupled since the meta learner is responsible for learning inductive biases that the base learner can exploit for future tasks.

The key technical insight of this work is to bound the generalization error at the two levels (base and meta) using two *different* generalization theory frameworks that each are particularly well-suited for addressing the specific challenges of generalization. At the base level, we utilize the fact that a learning algorithm that exhibits uniform stability [14, 15] also generalizes well in expectation (see Section 4.1 for a formal statement). Intuitively, uniform stability quantifies the sensitivity of the output of a learning algorithm to changes in the training dataset. As demonstrated by [29], limiting the number of training epochs of a gradient-based learning algorithm leads to uniform stability. In other words, a gradient-based algorithm that *learns quickly* is stable. Since the goal of meta-learning is *precisely* to train the base learner to learn quickly, we posit that generalization bounds based on stability are particularly well-suited to bounding the generalization error at the base level. At the meta level, we employ a generalization bound based on *Probably Approximately Correct (PAC)-Bayes* theory. Originally developed two decades ago [43, 38], there has been a recent resurgence of interest in PAC-Bayes due to its ability to provide strong generalization guarantees for neural networks [24, 8, 54]. Intuitively, the challenge of generalization at the meta level (i.e., generalizing to new tasks) is similar to the challenge of generalizing to new data in the standard supervised learning setting. In both cases, one must prevent over-fitting to the particular tasks/data that have been seen during meta-training/training. Thus, the strong empirical performance of PAC-Bayes theory in supervised learning problems makes it a promising candidate for bounding the generalization error at the meta level.

**Contributions.** The primary contributions of this work are the following. First, we leverage the insights above in order to develop a novel generalization bound for gradient-based meta-learning using uniform stability and PAC-Bayes theory (Theorem 3). Second, we develop a regularization scheme for MAML [25] that explicitly minimizes the derived bound (Algorithm 1). We refer to the resulting approach as *PAC-BUS* since it combines PAC-Bayes and Uniform Stability to derive generalization guarantees for meta-learning. Third, we demonstrate our approach on two meta-learning problems: (i) a toy non-convex classification problem on the unit-ball (Section 5.1), and (ii) the *Mini-Wiki* benchmark introduced in [34] (Section 5.2). Even in these relatively small-scale settings, we demonstrate that recently-developed generalization frameworks for meta-learning provide either near-vacuous or loose bounds, while PAC-BUS provides significantly stronger bounds. Fourth, we demonstrate our approach in larger-scale settings where it remains challenging to obtain non-vacuous bounds (for our approach as well as others). Here, we propose a practical regularization scheme which re-weights the terms in the rigorously-derived PAC-BUS upper bound (*PAC-BUS(H)*; Algorithm 3 in the appendix). Recent work [77] introduces a challenging variant of the *Omniglot* benchmark [35] which highlights and tackles challenges with *memorization* in meta-learning. We show that PAC-BUS(H) is able to prevent memorization on this variant (Section 5.3).

## 2   Problem formulation

**Samples, tasks, and datasets.** Formally, consider the setting where we have an unknown meta distribution $P_t$ over tasks (roughly, "tasks" correspond to different, but potentially related, learning problems). A sampled task $t \sim P_t$ induces an (unknown) distribution $P_{z|t}$ over sample space $\mathcal{Z}$. We assume that all sampling is independent and identically distributed (i.i.d.). Note that the sample space $\mathcal{Z}$ is shared between tasks, but the distribution $P_{z|t}$ may be different. We then sample within-task samples $z \sim P_{z|t}$ and within-task datasets $S = \{z_1, z_2, \ldots, z_m\} \sim P_{z|t}^m$. We assume that each sample $z$ has a single corresponding label $o(z)$, where the function $o$ is an oracle which outputs the correct label of $z$. At meta-training time, we assume access to $l$ datasets, which we call $\mathbf{S} = \{S_1, S_2, \ldots, S_l\}$. Each dataset $S_i$ in $\mathbf{S}$ is drawn by first selecting a task $t_i$ from $P_t$, and then drawing $S_i \sim P_{z|t_i}^m$.

**Hypotheses and losses.** Let $h$ denote a hypothesis and $L(h, z)$ be the loss incurred by hypothesis $h$ on sample $z$. The loss is computed by comparing $h(z)$ with the true label $o(z)$. For simplicity, we assume that there is no noise on the labels; we can thus assume that all loss functions have access to

the label oracle function $o$ and thus the loss depends only on hypothesis $h$ and sample $z$. We note that this assumption is not required for our analysis and is made for the ease of exposition. Overloading the notation, we let $L(h, P_{z|t}) := \mathbb{E}_{z \sim P_{z|t}} L(h, z)$ and $\widehat{L}(h, S) := \frac{1}{|S|} \sum_{i=1}^{|S|} L(h, z_i)$.

**Meta-learning.** As with model-agnostic meta-learning (MAML) [25], we let meta parameters $\theta \in \mathbb{R}^{n_\theta}$ correspond to an initialization of the base learner's hypothesis. Let $h_\theta$ be the $\theta$-initialized hypothesis. Generally, the initialization $\theta$ is learned from the multiple datasets we have access to at meta-training time. In this work, we will learn a *distribution* $P_\theta$ over initializations so that we can use bounds from the PAC-Bayes framework. At test time, a new task $t \sim P_t$ is sampled and we are provided with a new dataset $S \sim P_{z|t}^m$. The base learner uses an algorithm $A$ (e.g., gradient descent), the dataset $S$, and the initialization $\theta \sim P_\theta$ in order to fine-tune the hypothesis and perform well on future samples drawn from $P_{z|t}$. We denote the base learner's updated hypothesis by $h_{A(\theta, S)}$. More formally, our goal is to learn a distribution $P_\theta$ with the following objective:

$$\min_{P_\theta} \mathcal{L}(P_\theta, P_t) := \min_{P_\theta} \mathbb{E}_{t \sim P_t} \mathbb{E}_{S \sim P_{z|t}^m} \mathbb{E}_{\theta \sim P_\theta} L(h_{A(\theta, S)}, P_{z|t}). \tag{1}$$

We are particularly interested in the the few-shot learning case, where the number of samples which the base learner can use to adapt is small. A common technique to improve test performance in the few-shot learning case is to allow for validation data at meta-training time. Thus, in addition to a generalization guarantee on meta-learning without validation data, we will derive a bound when allowing for the use of validation data $S_{\text{va}} \sim P_{z|t}^n$ during meta-training.

## 3 Related work

**Meta-learning.** Meta-learning is a well-studied technique for exploiting similarities between learning tasks [62, 70]. Often used to reduce the need for large amounts of training data, a number of approaches for meta-learning have been explored over decades [11, 13, 16, 31, 72, 61, 55, 32]. Recently, methods based on model-agnostic meta-learning (MAML) [25] have demonstrated strong performance across different application domains and benchmarks such as *Omniglot* [35] and *Mini-ImageNet* [74]. These methods operate by optimizing a set of initial parameters that can be quickly fine-tuned via gradient descent on a new task. The approaches mentioned above typically do not provide any generalization guarantees, and none of them compute explicit numerical bounds on generalization performance. Our approach has the structure of gradient-based meta-learning while providing guarantees on generalization.

**Generalization bounds for supervised learning.** Multiple frameworks have been developed for providing generalization guarantees in the classical supervised learning setting. Early breakthroughs include Vapnik-Chervonenkis (VC) theory [71, 6], Rademacher complexity [65], and the minimum description length principle [12, 56, 36]. More recent frameworks include algorithmic stability bounds [14, 19, 29, 57, 1] and PAC-Bayes theory [67, 43, 64]. The connection between stability and learnability has been established in [66, 73, 29], and suggests that algorithmic stability bounds are a strong choice of generalization framework. PAC-Bayes theory in particular provides some of the tightest known generalization bounds for classical supervised learning approaches such as support vector machines [64, 38, 26, 57, 4, 48]. Since its development, researchers have continued to tighten [38, 42, 54] and generalize the framework [17, 18, 59]. Exciting recent results [24, 45, 46, 10, 8, 54] have demonstrated the promise of PAC-Bayes to provide strong generalization bounds for neural networks on supervised learning problems (see [33] for a recent review of generalization bounds for neural networks). It is also possible to combine frameworks such as PAC-Bayes and uniform stability to derive bounds for supervised learning [39]. We will use these two frameworks to bound generalization in the two levels of meta-learning. In contrast to the standard supervised learning setting, generalization bounds for meta-learning are less common and remain loose.

**Generalization bounds for meta-learning.** As described in Section 1, meta-learning bounds must account for two "levels" of generalization (base level and meta level). The approach presented in [41] utilizes algorithmic stability bounds at both levels. However, this requires both meta and base learners to be uniformly stable. This is a strong requirement that is challenging to ensure at the meta level. Another recent method, known as follow-the-meta-regularized-leader (FMRL) [34], provides guarantees for a regularized meta-learning version of the follow-the-leader (FTL) method for online learning, see e.g. [30]. The generalization bounds provided are derived from the application of online-to-batch techniques [3, 22]. A regret bound for meta-learning using an aggregation technique at the meta-level and an algorithm with a uniform generalization bound at the base level is provided

in [3]. The techniques mentioned do not present an algorithm which makes use of validation data (in contrast to our approach). Using validation data (i.e., held-out data) is a common technique for improving performance in meta-learning and is particularly important for the few-shot learning case.

Another method for deriving a generalization bound on meta-learning is to use PAC-Bayes bounds at both the base and meta levels [52, 53]. In [5], generalization bounds based on such a framework are provided along with practical optimization techniques. However, the method requires one to maintain distributions over distributions of initializations, which can result in large computation times during training and makes it extremely challenging to numerically compute the bound. Moreover, the approach also does not allow one to incorporate validation data to improve the bound. Recent work has made progress on some of these challenges. In [60], the computational efficiency of training is improved but the challenges associated with numerically computing the generalization bound or incorporating validation data are not addressed. State-of-the-art work tightens the two-level PAC-Bayes guarantee, addresses computation times for training and evaluation of the bound, and allows for validation data [77]. However, all of the two-level PAC-Bayes bounds require a separate PAC-Bayes bound for each task, and thus a potentially loose union bound.

We present a framework which, to our knowledge, is the first to combine algorithmic stability and PAC-Bayes bounds (at the base- and meta- levels respectively) in order to derive a meta-learning algorithm with associated generalization guarantees. As outlined in Section 1, we believe that the algorithmic stability and PAC-Bayes frameworks are particularly well-suited to tackling the specific challenges of generalization at the different levels. We also highlight that *none* of the approaches mentioned above report numerical values for generalization bounds, even for relatively simple problems. Here, we empirically demonstrate that prior approaches tend to provide either near-vacuous or loose bounds even in relatively small-scale settings while our proposed method provides significantly stronger bounds.

## 4 Generalization bound on meta-learning

We use two different frameworks for the two levels of generalization required in a meta-learning bound. We utilize the PAC-Bayes framework to bound the expected training loss on future tasks, and uniform stability bounds to argue that if we have a low training loss when using a uniformly stable algorithm, then we achieve a low test loss. The following section will introduce these frameworks independently. We then present the overall meta-learning bound and associated algorithm to find a distribution over initialization parameters (i.e., meta parameters) that minimizes the upper bound.

### 4.1 Preliminaries: two generalization frameworks

#### 4.1.1 Uniform stability

Let $S = \{z_1, z_2, \ldots, z_m\} \in \mathcal{Z}^m$ be a set of $m$ elements of $\mathcal{Z}$. Let $S^i = \{z_1, \ldots, z_{i-1}, z_i', z_{i+1}, \ldots, z_m\}$ be identical to dataset $S$ except that the $i^{th}$ sample $z_i$ is replaced by some $z_i' \in \mathcal{Z}$. Note that our analysis can be extended to allow for losses bounded by some finite $M$, but we work with losses bounded within $[0, 1]$ for the sake of simplicity. With these precursors, we define an analogous notion of *uniform stability* to [29, Definition 2.1] for deterministic algorithms $A$ and distributions $P_\theta$ over initializations.[1]

**Definition 1** (Uniform Stability) *A deterministic algorithm $A$ has $\beta > 0$ uniform stability with respect to loss $L$ if $\forall z \in \mathcal{Z}$, $\forall S \in \mathcal{Z}^m$, $\forall i \in \{1, \ldots, m\}$, and all distributions $P_\theta$ over initializations, the following holds:*

$$\mathbb{E}_{\theta \sim P_\theta} |L(h_{A(\theta, S)}, z) - L(h_{A(\theta, S^i)}, z)| \leq \beta. \tag{2}$$

*We define $\beta_{\text{US}}$ as the minimal such $\beta$.*

In this work, we will bound $\beta_{\text{US}}$ as a function of the algorithm, form of the loss, and number of samples that the algorithm uses (See Appendix A.4 for further details on the bounds on $\beta_{\text{US}}$ for our setup). We then establish a relationship between uniform stability and generalization in expectation. The following is adapted from [29, Theorem 2.2] for the notion of uniform stability presented in Definition 1:

---

[1]We use deterministic algorithms to avoid excess computation when calculating the provided meta-learning upper bounds. See Appendix A.4 for further details.

**Theorem 1** (Algorithmic Stability Generalization in Expectation) *Fix a task $t \sim P_t$. The following inequality holds for hypothesis $h_{A(\theta,S)}$ learned using $\beta_{\mathrm{US}}$ uniformly stable algorithm $A$ with respect to loss $L$:*

$$\mathop{\mathbb{E}}_{S \sim P_{z|t}^m} \mathop{\mathbb{E}}_{\theta \sim P_\theta} L(h_{A(\theta,S)}, P_{z|t}) \leq \mathop{\mathbb{E}}_{S \sim P_{z|t}^m} \mathop{\mathbb{E}}_{\theta \sim P_\theta} \widehat{L}(h_{A(\theta,S)}, S) + \beta_{\mathrm{US}}. \tag{3}$$

*Proof.* The proof is similar to the one presented for [29, Theorem 2.2] and is presented in Appendix A.1. $\qquad\square$

### 4.1.2 PAC-Bayes theory

For the meta-level bound, we make use of the PAC-Bayes generalization bound introduced in [43]. Note that other PAC-Bayes bounds such as the quadratic variant [58] and PAC-Bayes-$\lambda$ variant [69] may be used and substituted in the following analysis. We first present a general version of the PAC-Bayes bound and then specialize it to our meta-learning setting in Section 4.2. Let $f(\theta, s)$ be an arbitrary loss function which only depends on parameters $\theta$ and the sample $s$ which has been drawn from an arbitrary distribution $P_s$. The following bound is a tightened version of the bound presented in [43] for when $l \geq 8$.

**Theorem 2** (PAC-Bayes Generalization Bound [40]) *For any data-independent prior distribution $P_{\theta,0}$ over $\theta$, some loss function $f$ where $0 \leq f(\theta, s) \leq 1, \forall\, s, \forall\, \theta,\, l \geq 8$, and $\delta \in (0, 1)$, with probability at least $1 - \delta$ over a sampling of $\{s_1, s_2, \ldots, s_l\} \sim P_s^l$, the following holds simultaneously for all distributions $P_\theta$ over $\theta$:*

$$\mathop{\mathbb{E}}_{s \sim P_s} \mathop{\mathbb{E}}_{\theta \sim P_\theta} f(\theta, s) \leq \frac{1}{l} \sum_{i=1}^{l} \mathop{\mathbb{E}}_{\theta \sim P_\theta} f(\theta, s_i) + R_{\mathrm{PAC-B}}(P_\theta, P_{\theta,0}, \delta, l), \tag{4}$$

*where the PAC-Bayes "regularizer" term is defined as follows*

$$R_{\mathrm{PAC-B}}(P_\theta, P_{\theta,0}, \delta, l) := \sqrt{\frac{D_{\mathrm{KL}}(P_\theta \| P_{\theta,0}) + \ln \frac{2\sqrt{l}}{\delta}}{2l}}, \tag{5}$$

*and $D_{\mathrm{KL}}$ is the Kullback-Leibler (KL) divergence.*

## 4.2 Meta-learning bound

In order to obtain a generalization guarantee for meta-learning, we utilize the two frameworks above. We first specialize the PAC-Bayes bound in Theorem 2 to bound the expected training loss on future tasks. We then utilize Theorem 1 to demonstrate that if we have a low expected training loss when using a uniformly stable algorithm, then we achieve a low expected test loss. These two steps allow us to combine the generalization frameworks above to derive an upper bound on (1) which can be computed with known quantities. With the following assumption, the resulting generalization bound is presented in Theorem 3.

**Assumption 1** (Bounded loss.) *The loss function $L$ is bounded: $0 \leq L(h, z) \leq 1$ for any $h$ in the hypothesis space for the given problem, and any $z$ in the sample space.*

**Theorem 3** (Meta-Learning Generalization Guarantee) *For hypotheses $h_{A(\theta,S)}$ learned with $\beta_{\mathrm{US}}$ uniformly stable algorithm $A$, data-independent prior $P_{\theta,0}$ over initializations $\theta$, loss $L$ which satisfies Assumption 1, $l \geq 8$, and $\delta \in (0, 1)$, with probability at least $1 - \delta$ over a sampling of the meta-training dataset $\mathbf{S} \sim P_S^l$, the following holds simultaneously for all distributions $P_\theta$ over $\theta$:*

$$\mathcal{L}(P_\theta, P_t) \leq \frac{1}{l} \sum_{i=1}^{l} \mathop{\mathbb{E}}_{\theta \sim P_\theta} \widehat{L}(h_{A(\theta,S_i)}, S_i) + R_{\mathrm{PAC-B}}(P_\theta, P_{\theta,0}, \delta, l) + \beta_{\mathrm{US}}. \tag{6}$$

*Proof.* The proof can be split into three steps:
**Step 1.**
Let $P_s$ in Theorem 2 be the marginal distribution $P_S$ over datasets of size $m$ (see Appendix A.2 for details) and note that sampling $S \sim P_S$ is equivalent to first sampling $t \sim P_t$ and then sampling $S \sim P_{z|t}^m$. Additionally let $f(\theta, S) := \widehat{L}(h_{A(\theta,S)}, S)$ where $A(\theta, S)$ is any deterministic algorithm.

Plugging in these definitions into Inequality (4) results in the following inequality which holds under the same assumptions as Theorem 2, and with probability at least $1 - \delta$ over the sampling of $\mathbf{S} \sim P_S^l$:

$$\mathop{\mathbb{E}}_{S \sim P_S} \mathop{\mathbb{E}}_{\theta \sim P_\theta} \widehat{L}(h_{A(\theta,S)}, S) = \mathop{\mathbb{E}}_{t \sim P_t} \mathop{\mathbb{E}}_{S \sim P_{z|t}^m} \mathop{\mathbb{E}}_{\theta \sim P_\theta} \widehat{L}(h_{A(\theta,S)}, S)$$

$$\leq \frac{1}{l} \sum_{i=1}^l \mathop{\mathbb{E}}_{\theta \sim P_\theta} \widehat{L}(h_{A(\theta,S_i)}, S_i) + R_{\mathrm{PAC-B}}(P_\theta, P_{\theta,0}, \delta, l). \qquad (7)$$

**Step 2.**
Now assume that algorithm $A$ is $\beta_{\mathrm{US}}$ uniformly stable. For a fixed task $t \sim P_t$ we have the following by Theorem 1:

$$\mathop{\mathbb{E}}_{S \sim P_{z|t}^m} \mathop{\mathbb{E}}_{\theta \sim P_\theta} L(h_{A(\theta,S)}, P_{z|t}) \leq \mathop{\mathbb{E}}_{S \sim P_{z|t}^m} \mathop{\mathbb{E}}_{\theta \sim P_\theta} \widehat{L}(h_{A(\theta,S)}, S) + \beta_{\mathrm{US}}.$$

Take the expectation over $t \sim P_t$. We then have:

$$\mathop{\mathbb{E}}_{t \sim P_t} \mathop{\mathbb{E}}_{S \sim P_{z|t}^m} \mathop{\mathbb{E}}_{\theta \sim P_\theta} L(h_{A(\theta,S)}, P_{z|t}) \leq \mathop{\mathbb{E}}_{t \sim P_t} \mathop{\mathbb{E}}_{S \sim P_{z|t}^m} \mathop{\mathbb{E}}_{\theta \sim P_\theta} \widehat{L}(h_{A(\theta,S)}, S) + \beta_{\mathrm{US}}, \qquad (8)$$

since $\mathbb{E}_{t \sim P_t} \beta_{\mathrm{US}} = \beta_{\mathrm{US}}$. This establishes a bound on the true expected loss for a new task after running algorithm A on a training dataset corresponding to the new task.
**Step 3.**
Note that (7) provides an upper bound on the first term of the RHS of (8) when algorithm $A$ is $\beta_{\mathrm{US}}$ uniformly stable. Thus we have the following by plugging (7) in the RHS of (8):
Under the same assumptions as both Theorems 1 and 2, and with probability at least $1 - \delta$ over the sampling of $\mathbf{S} \sim P_S^l$:

$$\mathop{\mathbb{E}}_{t \sim P_t} \mathop{\mathbb{E}}_{S \sim P_{z|t}^m} \mathop{\mathbb{E}}_{\theta \sim P_\theta} L(h_{A(\theta,S)}, P_{z|t}) \leq \frac{1}{l} \sum_{i=1}^l \mathop{\mathbb{E}}_{\theta \sim P_\theta} \widehat{L}(h_{A(\theta,S_i)}, S_i) + R_{\mathrm{PAC-B}}(P_\theta, P_{\theta,0}, \delta, l) + \beta_{\mathrm{US}},$$

completing the proof. $\qquad \square$

Theorem 3 is presented for any distributions $P_\theta$ and $P_{\theta,0}$ over initializations. However, in practice we will use multivariate Gaussian distributions for both. The specialization of Theorem 3 to Gaussian distributions is provided in Appendix A.3.1. Next, we allow for validation data $S_{\mathrm{va}} \sim P_{z|t}^n$ at meta-training time so that the bound is more suited to the few-shot learning case. We compute the upper bound using the evaluation data $S_{\mathrm{ev}} = \{S, S_{\mathrm{va}}\}$ sampled from the marginal distribution $P_{S_{\mathrm{ev}}}$ over datasets of size $m + n$. However, we still only require $m$ samples at meta-test time; see Appendix A.3.2 for the derivation. Note that the training data $S$ is often excluded from the data used to update the meta-learner. However, this is necessary for our approach to obtain a guarantee on few-shot learning performance. The result is a guarantee with high probability over a sampling of $\mathbf{S}_{\mathrm{ev}} \sim P_{S_{\mathrm{ev}}}^l$:

$$\mathcal{L}(P_\theta, P_t) \leq \frac{1}{l} \sum_{i=1}^l \mathop{\mathbb{E}}_{\theta \sim P_\theta} \widehat{L}(h_{A(\theta,S_i)}, S_{\mathrm{ev},i}) + R_{\mathrm{PAC-B}}(P_\theta, P_{\theta,0}, \delta, l) + \frac{m \beta_{\mathrm{US}}}{m + n}. \qquad (9)$$

### 4.3 PAC-BUS algorithm

Recall that we aim to find a distribution $P_\theta$ over initializations that minimizes $\mathcal{L}(P_\theta, P_t)$ as stated in Equation (1). We cannot minimize $\mathcal{L}(P_\theta, P_t)$ directly due to the expectations taken over unknown distributions $P_t$ and $P_{z|t}$ for sampled task $t$, but we may indirectly minimize it by minimizing the upper bounds in Inequalities (6) or (9).

Computing the upper bound requires evaluating an expectation taken over $\theta \sim P_\theta$. In general, this is intractable. However, we aim to minimize this upper bound to provide the tightest guarantee possible. Similar to the method in [24], we use an unbiased estimator of $\mathbb{E}_{\theta \sim P_\theta} L(h_{A(\theta,S)}, \cdot)$. Let $P_\theta$ be a multivariate Gaussian distribution over initializations $\theta$ with mean $\mu$ and covariance $\mathrm{diag}(s)$; thus $P_\theta = \mathcal{N}(\mu, \mathrm{diag}(s))$ and $P_{\theta,0} = \mathcal{N}(\mu_0, \mathrm{diag}(s_0))$. Further, let $\psi := (\mu, \log(s))$, and use the shorthand $\mathcal{N}_{\psi_0}$ for the prior and $\mathcal{N}_\psi$ for the posterior distribution over initializations. We use the following estimator of $\mathbb{E}_{\theta \sim P_\theta} L(h_{A(\theta,S)}, \cdot)$:

$$L(h_{A(\theta,S)}, \cdot), \quad \theta \sim \mathcal{N}_\psi. \qquad (10)$$

**Algorithm 1** PAC-BUS: meta-learning via PAC-Bayes and Uniform Stability
___

**Input**: Fixed prior distribution $\mathcal{N}_{\psi_0}$ over initializations
**Input**: $\beta_{\text{US}}$ uniformly stable Algorithm $A$
**Input**: Meta-training dataset **S**, learning rate $\gamma$
**Initialize**: $\psi \leftarrow \psi_0$
**Output**: Optimized $\psi^*$
$B(\psi, \theta'_1, \theta'_2 \ldots, \theta'_l) := \frac{1}{l} \sum_{i=1}^l \widehat{L}(h_{\theta'_i}, S_i) + R_{\text{PAC-B}}(\mathcal{N}_\psi, \mathcal{N}_{\psi_0}, \delta, l) + \beta_{\text{US}}$
**while** not converged **do**
    Sample $\theta \sim \mathcal{N}_\psi$
    **for** $i = 1$ **to** $l$ **do**
        $\theta'_i \leftarrow A(\theta, S_i)$
    **end for**
    $\psi \leftarrow \psi - \gamma \nabla_\psi B(\psi, \theta'_1, \theta'_2 \ldots, \theta'_l)$
**end while**
___

We present the resulting training technique in Algorithm 1. This algorithm can be used to learn a distribution over initializations that minimizes the upper bound presented in Theorem 3 and its specializations. This is presented for the case when $A$ is $\beta_{\text{US}}$ uniformly stable for some $\beta_{\text{US}}$. For gradient-based algorithms, the learning rate $\alpha$ often appears directly in the bound for $\beta_{\text{US}}$ [29]. Thus it is potentially beneficial to update $\alpha$ as well. We present Algorithm 1 without learning the learning rate. To meta-learn the learning rate, we can augment $\psi_0$ to include a parameterization of a prior distribution over learning rates and update it using the same gradient step presented in 1 for $\psi$.

Determining the gradient of $B(\psi, \theta'_1, \theta'_2 \ldots, \theta'_l)$ with respect to $\psi$ requires computing the Hessian of the loss function if algorithm $A(\theta, S)$ uses a gradient update to compute $\theta'_i$. First order approximations often perform similarly to the second-order meta-learning techniques [27, 25, 47], and can be used to speed up the training. Additionally, Algorithm 1 can be modified to use mini-batches of tasks instead of all tasks in the meta update to improve training times; we present an algorithm which uses mini-batches of tasks in Appendix A.5.1.

In practice, we are interested in algorithms such as stochastic gradient descent (SGD) and gradient descent (GD) for the base learner. We can obtain bounds on the uniform stability constant $\beta_{\text{US}}$ when using gradient methods with the results from [29]. See Appendix A.4 for details on the $\beta_{\text{US}}$ bounds we use in this work. With a bound on $\beta_{\text{US}}$, we can calculate all the terms in $B(\psi, \theta'_1, \theta'_2 \ldots, \theta'_l)$ and use Algorithm 1 to minimize the meta-learning upper bound. When evaluating the upper bound, we use the sample convergence bound [37, 24] to upper bound the expectation taken over $\theta \sim P_\theta$. See Appendix A.6 for details.

## 5 Examples

We demonstrate our approach on three examples below. All examples we provide are few-shot meta-learning problems. To adapt at the base level, $m$ examples from each class are given for an "$m$-shot" learning problem. If applicable, $n$ samples can be given as validation data for each task during the meta-training step. In the first two examples, our primary goal is to demonstrate the tightness of our generalization bounds compared to other meta-learning bounds. We also present empirical test performance on held-out data; however, we emphasize that the focus of our work is to obtain improved generalization guarantees (and not necessarily to improve empirical test performance). In the third example, we present an algorithm that is motivated by our theoretical framework and demonstrate its ability to improve empirical performance on a challenging task. All the code required to run the following examples is available at `https://github.com/irom-lab/PAC-BUS`.

### 5.1 Example: classification on the unit ball

We evaluate the tightness of the generalization bound in Equation (9) on a toy two-class classification problem where the sample space $\mathcal{Z}$ is the unit ball $B^2(0, 1)$ in two dimensions with radius 1 and centered at the origin. Data points for each task are sampled from $P_{z|t}$, where a task corresponds to a particular concept which labels the data as $(+)$ if within $B^2(c_t, r_t)$ and $(-)$ otherwise. Center $c_t$ is sampled uniformly from the $y \geq 0$ semi-ball $B^2_{y \geq 0}(0, 0.4)$ of radius 0.4. The radius $r_t$ is then sampled uniformly from $[0.1, 1 - \|c_t\|]$. Notably, the decision boundary between classes is nonlinear. Thus, generalization bounds which rely on convex losses (such as [34]) will have difficulty with

Table 1: We present the generalization bounds (for $\delta = 0.01$) provided by each method if applicable, and use the sample convergence bound [37] for MR-MAML, and PAC-BUS, but not MLAP-M.[2] Note that for these methods, we specifically minimize their respective meta-learning bounds. We also report the meta-test loss (the softmax activated cross-entropy loss – $CEL_s$) for all methods. We present the mean and standard deviation after 5 trials. We highlight that our approach provides the strongest generalization guarantee.

| Classification on Ball | MAML [25] | MLAP-M [5] | MR-MAML [77] | PAC-BUS (ours) |
|---|---|---|---|---|
| Bound ↓ | None | $1.0538 \pm 0.0012^2$ | $0.3422 \pm 0.0006$ | $\mathbf{0.2213 \pm 0.0012}$ |
| Test Loss ↓ | $0.1701 \pm 0.0070$ | $0.1645 \pm 0.0045$ | $\mathbf{0.1584 \pm 0.0012}$ | $0.1657 \pm 0.0014$ |

providing guarantees for networks that perform well. We choose the softmax-activated cross-entropy loss, $CEL_s$, as the loss function. Before running Algorithm 1, we address a few technical challenges that arise from Assumption 1 as well as computing $c_L$ and $c_S$. We address these in Appendix A.5. []

We then apply Algorithm 1 using the few-shot learning bound in Inequality (9). We present the guarantee on the meta-test loss associated with each training method in Table 1. In addition, we present the average meta-test loss after training with 10 samples. We compare our bounds and empirical performance with the meta-learning by adjusting priors (MLAP) technique [5] and the meta-regularized MAML (MR-MAML) technique [77]. All methods are given held-out data to learn a prior before minimizing their respective upper bounds (see Appendix A.11.1 for further details on the prior training step). Additionally, since all bounds require the loss to be within $[0, 1]$, networks $N$ are constrained such that the Frobenius norm of the output is bounded by $r$, i.e., $\|N(z)\|_F \le r$. We compare the aforementioned methods' meta-test loss to MAML with weights constrained in the same manner (note that MAML does not provide a guarantee). Upper bounds which use the PAC-Bayes framework are computed with many evaluations from the posterior distribution. This allows us to apply the sample convergence bound [37] (as in Equation (35) for our bound) unless otherwise noted.

We find that PAC-BUS provides a significantly stronger guarantee compared with the other methods. Note that the guarantee provided by MLAP-M [5] is vacuous because the meta-test loss is bounded between 0 and 1, while the guarantee is above 1.

## 5.2   Example: Mini-Wiki

Next, we present results on the *Mini-Wiki* benchmark introduced in [34]. This is derived from the Wiki3029 dataset presented in [9]. The dataset is comprised of 4-class, $m$-shot learning tasks with sample space $\mathcal{Z} = \{z \in \mathbb{R}^d \mid \|z\|_2 = 1\}$. Sentences from various Wikipedia articles are passed through the continuous-bag-of-words GloVe embedding [51] into dimension d = 50 to generate samples. For this learning task, we use a $k$-class version of $CEL_s$ and logistic regression. Since this example is convex, we can use GD and bound $\beta_{US}$ with Theorem 4 in the appendix [29]. We keep the loss bounded by constraining the network $\|N(z)\|_F \le r$ and scale the loss as in the previous example. The tightness of the bounds on $c_L$ and $c_S$ affected the upper bound in Inequality (9) more than in the previous example, so we bound them as tightly as possible. See Appendix A.9 for the calculations.

We apply Algorithm 1 using the bound which allows for validation data, Inequality (9), to learn on 4-way *Mini-Wiki* $m = \{1, 3, 5\}$-shot. The results are presented in Table 2. We compare our results with the FMRL variant which provides a guarantee [34], follow-the-last-iterate (FLI)-Batch, and with MR-MAML [77]. FLI-Batch does not require bounded losses explicitly, but requires that the parameters of the network lie within a ball of radius $r$. For the logistic regression used in the example, this is equivalent to $\|N(z)\|_F \le r$. Thus, we scale the loss and use the same $r$ for each method to provide a fair comparison. We also show the results of training with MAML constrained in the same way for reference. Each method is given the same amount of held-out data for training a prior (see Appendix A.11.2 for further details on training the prior).

---

[2]Due to high computation times associated with estimating the MLAP upper bound, this value is not computed with the sample convergence bound as the other upper bounds are. Thus, the value presented does not carry a guarantee, but would be similar if computed with the sample convergence bound. The value is shown to give a qualitative sense of the guarantee.

Table 2: We compare the generalization bounds (for $\delta = 0.01$) provided by each method where applicable and use the sample convergence bound for MR-MAML and PAC-BUS. Since we specifically minimize these methods' upper bounds, we can fairly compare the relative tightness of each bound. We also report the meta-test loss ($\text{CEL}_s$) for each method for exposition. We report the mean and standard deviation after 5 trials. We highlight that our approach provides the strongest guarantee.

| 4-Way *Mini-Wiki* | 1-shot ↓ | 3-shot ↓ | 5-shot ↓ |
|---|---|---|---|
| FLI-Batch Bound [34] | $0.6638 \pm 0.0011$ | $0.6366 \pm 0.0006$ | $0.6343 \pm 0.0014$ |
| MR-MAML Bound [77] | $0.7400 \pm 0.0003$ | $0.7312 \pm 0.0003$ | $0.7283 \pm 0.0005$ |
| PAC-BUS Bound (ours) | $\mathbf{0.4999 \pm 0.0003}$ | $\mathbf{0.5058 \pm 0.0002}$ | $\mathbf{0.5101 \pm 0.0002}$ |
| MAML [25] | $\mathbf{0.3916 \pm 0.0009}$ | $\mathbf{0.3868 \pm 0.0005}$ | $\mathbf{0.3883 \pm 0.0005}$ |
| FLI-Batch [34] | $0.4091 \pm 0.0008$ | $0.4078 \pm 0.0005$ | $0.4097 \pm 0.0012$ |
| MR-MAML [77] | $0.3922 \pm 0.0009$ | $0.3869 \pm 0.0003$ | $0.3884 \pm 0.0005$ |
| PAC-BUS (ours) | $0.3922 \pm 0.0009$ | $0.3878 \pm 0.0003$ | $0.3895 \pm 0.0005$ |

As in the previous example, PAC-BUS provides a significantly tighter guarantee than the other methods (Table 2). We see similar empirical meta-test loss for MAML [25], MR-MAML [77], and PAC-BUS with slightly higher loss for FLI-Batch [34]. In addition, we computed the meta-test accuracy as the percentage of correctly classified sentences. See Table 4 in Section A.11.2 for these results along with other experimental details.

## 5.3 Example: memorizable Omniglot

We have demonstrated the ability of our approach to provide strong generalization guarantees for meta-learning in the settings above. We now consider a more complex setting where we are unable to obtain strong guarantees. In this example, we employ a learning heuristic based on the PAC-BUS upper bound, *PAC-BUS(H)*; see Appendix A.5.2 for the details and the Algorithm. We relax Assumption 1 and no longer constrain the network as in previous sections. Instead, we maintain and update estimates of the Lipschitz and smoothness constants of the network, using [68], and incorporate them into the uniform stability regularizer term, $\beta_{\text{US}}$. We then scale each regularizer term (i.e., $R_{\text{PAC}-\text{B}}(P_\theta, P_{\theta,0}, \delta, l)$ and $\beta_{\text{US}}$) by hyper-parameters $\lambda_1$ and $\lambda_2$ respectively. Analogous to the technique described in [77], we aim to incorporate the form of the theoretically-derived regularizer into the loss, without requiring it to be as restrictive during learning. The result is a regularizer that punishes large deviation from the prior $P_{\theta,0}$ and too much adaptation at the base-learning level.

We test our method on *Omniglot* [35] for 20-way, $m = \{1, 5\}$-shot classification in the non-mutually exclusive (NME) case [77]. In [77], the problem of memorization in meta-learning is explored and demonstrated with non-mutually exclusive learning problems. *NME Omniglot* corresponds to randomization of class labels for a task at test time only. This worsens the performance of any network that memorized class labels; see [77] for more details.[3] We compare our method to an analogous heuristic presented in [77], which also has a $D_{\text{KL}}(P_\theta \| P_{\theta,0})$ term in the loss. Thus, this heuristic (referred to as MR-MAML(W) [77]) regularizes the change in weights of the network. Additionally, we compare to the heuristic described in [34] (FLI-Online) which performs better in practice than the FLI-Batch method. We do not provide data for training a prior in this case since we do not aim to compute a bound in this example. We use standard MAML as a reference. See Table 3 for the results.

We see that MAML [25] and FLI-Online [34] do not prevent memorization on *NME Omniglot* [77]. This is especially apparent in the 1-shot learning case, where their performance suffers significantly due to this memorization. Both MR-MAML(W) [77] and PAC-BUS(H) prevent memorization, with PAC-BUS(H) outperforming MR-MAML(W). Note that PAC-BUS(H) outperforms MR-MAML(W) by a wider margin in the 1-shot case as compared with the 5-shot case. We believe this is due to the effectiveness of the uniform stability regularizer at the base level. MR-MAML(W) suffers more in the 1-shot case because over-adaptation is more likely with fewer within-task examples. ∥

## 6 Conclusion and discussion

We presented a novel generalization bound for gradient-based meta-learning: PAC-BUS. We use different generalization frameworks for tackling the distinct challenges of generalization at the two

---

[3]We use a slightly different task setup as the one in [77]; see Appendix A.11.3 for the details of our setup.

Table 3: We present the meta-test accuracy as a percentage on non-mutually-exclusive *Omniglot* [77]. In contrast to the previous examples, here we aim to achieve the best empirical performance for each method. In particular, this task compares each methods' ability to prevent memorization. We report the mean and standard deviation after 5 trials.

| 20-WAY *Omniglot* | NME 1-SHOT ↑ | NME 5-SHOT ↑ |
|---|---|---|
| MAML [25] | $23.4 \pm 2.2$ | $75.1 \pm 4.8$ |
| FLI-ONLINE [34] | $22.4 \pm 0.5$ | $39.1 \pm 0.5$ |
| MR-MAML(W) [77] | $84.2 \pm 2.2$ | $94.3 \pm 0.3$ |
| PAC-BUS(H) (OURS) | $\mathbf{87.9 \pm 0.5}$ | $\mathbf{95.0 \pm 0.9}$ |

levels of meta-learning. In particular, we employ uniform stability bounds and PAC-Bayes bounds at the base- and meta-learning levels respectively. On a toy non-convex problem and the *Mini-Wiki* meta-learning task [34], we provide significantly tighter generalization guarantees as compared to state-of-the-art meta-learning bounds while maintaining comparable empirical performance. To our knowledge, this work presents the first numerically-evaluated generalization guarantees associated with a proposed meta-learning bound. On memorizable *Omniglot* [35, 77], we show that a heuristic based on the PAC-BUS bound prevents memorization of class labels in contrast to MAML [25], and better performance than meta-regularized MAML [77]. We believe our framework is well suited to the few-shot learning problems for which we present empirical results, but our framework is potentially applicable to a broad range of different settings (e.g., reinforcement learning).

We note a few challenges with our method as motivation for future work. Our bound is vacuous on larger scale learning problems such as *Omniglot*. This is partially caused by a larger KL-divergence term in the PAC-Bayes bound when using deep convolutional networks (due to the increased dimensionality of the weight vector). In addition, we do not have a theoretical analysis on the convergence properties of the algorithms presented, so we must experimentally determine the number of samples required for tight bounds. In the results of Section 5.1 and 5.2, despite an improved bound over other methods, our method does not necessarily improve empirical test performance. We emphasize that our focus in this work was on deriving stronger generalization guarantees rather than improving empirical performance. However, obtaining approaches that provide both stronger guarantees and empirical performance is an important direction for future work.

Future work can also explore ways in which to incorporate tighter PAC-Bayes bounds or those with less restrictive assumptions. One interesting avenue is to extend PAC-BUS by using a PAC-Bayes bound for unbounded loss functions for the meta-generalization step (e.g. as presented in [28]). Another promising direction is to incorporate regularization on the weights of the network directly (e.g., $L_2$ regularization or gradient clipping) to create networks with smaller Lipschitz and smoothness constants. Additionally, it would be interesting to explore learning of the base-learner's algorithm while maintaining uniform stability. For example, one could parameterize a set of uniformly stable algorithms and learn a posterior distribution over the parameters.

**Broader impact.** The approach we present in this work aims to strengthen performance guarantees for gradient-based meta-learning. We believe that strong generalization guarantees in meta-learning, especially in the few-shot learning case, could lead to broader application of machine learning in real-world applications. One such example is for medical diagnosis, where abundant training data for certain diseases may be difficult to obtain. Another example on which poor performance is not an option is any safety critical robotic system, such as ones which involve human interaction.

Meta-learning methods typically require a lot of data and training time, and ours is not an exception. In our case, it took multiple weeks of computation time on Amazon Web Services (AWS) instances to train and compute all networks and results we present in this paper. This creates challenges with accessibility and energy usage.

### Acknowledgments

The authors are grateful to the anonymous reviewers for their valuable feedback and suggestions, and to Thomas Griffiths for helpful feedback on this work. The authors were supported by the Office of Naval Research [N00014-21-1-2803, N00014-18-1-2873], the NSF CAREER award [2044149], the Google Faculty Research Award, and the Amazon Research Award.

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
