# A  Appendix

## A.1  Proof of Theorem 1

**Theorem 1** (Algorithmic Stability Generalization in Expectation) *Fix a task $t \in P_t$. The following inequality holds for hypothesis $h_{A(\theta,S)}$ learned using $\beta_{\text{US}}$ uniformly stable algorithm $A$ with respect to loss $L$:*

$$\mathop{\mathbb{E}}_{S \sim P_{z|t}^m} \mathop{\mathbb{E}}_{\theta \sim P_\theta} L(h_{A(\theta,S)}, P_{z|t}) \leq \mathop{\mathbb{E}}_{S \sim P_{z|t}^m} \mathop{\mathbb{E}}_{\theta \sim P_\theta} \widehat{L}(h_{A(\theta,S)}, S) + \beta_{\text{US}}. \tag{11}$$

*Proof.* Let $S = \{z_1, z_2 \ldots, z_m\} \sim P_{z|t}^m$ and $S' = \{z_1', z_2' \ldots, z_m'\} \sim P_{z|t}^m$ be two independent random samples and let $S^i = \{z_1, \ldots, z_{i-1}, z_i', z_{i+1}, \ldots, z_m\}$ be identical to $S$ except with the $i^{\text{th}}$ sample replaced with $z_i'$. Fix a distribution $P_\theta$ over initializations. Consider the following

$$\mathop{\mathbb{E}}_{S \sim P_{z|t}^m} \mathop{\mathbb{E}}_{\theta \sim P_\theta} \widehat{L}(h_{A(\theta,S)}, S) = \mathop{\mathbb{E}}_{S \sim P_{z|t}^m} \mathop{\mathbb{E}}_{\theta \sim P_\theta} \left[ \frac{1}{m} \sum_{i=1}^m L(h_{A(\theta,S)}, z_i) \right] \tag{12}$$

$$= \mathop{\mathbb{E}}_{S \sim P_{z|t}^m} \mathop{\mathbb{E}}_{S' \sim P_{z|t}^m} \mathop{\mathbb{E}}_{\theta \sim P_\theta} \left[ \frac{1}{m} \sum_{i=1}^m L(h_{A(\theta,S_i)}, z_i) \right] \tag{13}$$

$$= \mathop{\mathbb{E}}_{S \sim P_{z|t}^m} \mathop{\mathbb{E}}_{S' \sim P_{z|t}^m} \mathop{\mathbb{E}}_{\theta \sim P_\theta} \left[ \frac{1}{m} \sum_{i=1}^m L(h_{A(\theta,S)}, z_i') \right] + \delta \tag{14}$$

$$= \mathop{\mathbb{E}}_{S \sim P_{z|t}^m} \mathop{\mathbb{E}}_{\theta \sim P_\theta} L(h_{A(\theta,S)}, P_{z|t}) + \delta \tag{15}$$

where

$$\delta = \mathop{\mathbb{E}}_{S \sim P_{z|t}^m} \mathop{\mathbb{E}}_{S' \sim P_{z|t}^m} \mathop{\mathbb{E}}_{\theta \sim P_\theta} \left[ \frac{1}{m} \sum_{i=1}^m L(h_{A(\theta,S_i)}, z_i') - \sum_{i=1}^m L(h_{A(\theta,S)}, z_i') \right]. \tag{16}$$

We bound $\delta$ with the supremum over datasets $S$ and $S'$ differing by a single sample

$$\delta \leq \sup_{S,S',z} \mathop{\mathbb{E}}_{\theta \sim P_\theta} \left[ L(h_{A(\theta,S)}, z) - L(h_{A(\theta,S)}, z) \right] \leq \beta_{\text{US}} \tag{17}$$

by Definition 1. $\qquad\qquad\qquad\qquad\qquad\qquad\qquad\qquad\qquad\qquad\qquad\qquad\qquad\qquad\qquad\square$

## A.2  Definition of Marginal Distribution $P_S$

In this section we formally define the marginal distribution $P_S$ which we make use of in the proof of Theorem 3. This is the distribution over datasets one obtains by first sampling a task $t$ from $P_t$, and then sampling a dataset $S$ from $P_{z|t}^m$. Consider the following equations (for simplicity, we use summations instead of integrals to compute expectations; $p(t)$ represents the probability of sampling $t$ and $p(S|t)$ is the probability of sampling $S$ given $t$):

$$\mathop{\mathbb{E}}_{t \sim P_t} \mathop{\mathbb{E}}_{S \sim P_{z|t}^m} \mathop{\mathbb{E}}_{\theta \sim P_\theta} f(\theta, S) = \sum_t p(t) \sum_S p(S|t) \mathop{\mathbb{E}}_{\theta \sim P_\theta} f(\theta, S) \tag{18}$$

$$= \sum_{t,S} p(t)p(S|t) \mathop{\mathbb{E}}_{\theta \sim P_\theta} f(\theta, S) \tag{19}$$

$$= \sum_{t,S} p(S,t) \mathop{\mathbb{E}}_{\theta \sim P_\theta} f(\theta, S) \tag{20}$$

$$= \sum_S \left( \mathop{\mathbb{E}}_{\theta \sim P_\theta} f(\theta, S) \underbrace{\sum_t p(S,t)}_{=p(S)} \right). \tag{21}$$

Here $p(S)$ corresponds to the marginal distribution over datasets $S$. Note that the last line above holds because $\mathbb{E}_{\theta \sim P_\theta} f(\theta, S)$ does not depend on $t$.

**Definition 2** (Marginal Distribution Over Datasets $S$) *Let $P_S := p(S)$ from above.*

Thus we have

$$\mathop{\mathbb{E}}_{t \sim P_t} \mathop{\mathbb{E}}_{S \sim P_{z|t}^m} \mathop{\mathbb{E}}_{\theta \sim P_\theta} f(\theta, S) = \sum_S p(S) \mathop{\mathbb{E}}_{\theta \sim P_\theta} f(\theta, S) = \mathop{\mathbb{E}}_{S \sim P_S} \mathop{\mathbb{E}}_{\theta \sim P_\theta} f(\theta, S). \tag{22}$$

## A.3 Specializing the Bound

### A.3.1 Meta-Learning Bound for Gaussian Distributions

In practice, the distribution $P_\theta$ over initializations will be a multivariate Gaussian distribution. Thus, in this section, we present a specialization of the bound for Gaussian distributions. Let $P_\theta$ have mean $\mu$ and covariance $\Sigma$; thus $P_\theta = \mathcal{N}(\mu, \Sigma)$ and analogously $P_{\theta,0} = \mathcal{N}(\mu_0, \Sigma_0)$. We can then apply the analytical form for the KL-divergence between two multivariate Gaussian distributions to the bound presented in Theorem 3. The result is the following bound holding under the same assumptions as Theorem 3:

$$\mathcal{L}(P_\theta, P_t) \le \frac{1}{l} \sum_{i=1}^l \mathop{\mathbb{E}}_{\theta \sim P_\theta} L(h_{A(\theta, S_i)}, S_i) + \beta_{\mathrm{US}}$$

$$+ \sqrt{\frac{(\mu - \mu_0)\Sigma_0^{-1}(\mu - \mu_0) + \ln \frac{|\Sigma_0|}{|\Sigma|} + \mathrm{tr}(\Sigma_0^{-1}\Sigma) - n_{\mathrm{dim}} + 2\ln \frac{2\sqrt{l}}{\delta}}{4l}}, \tag{23}$$

where $n_{\mathrm{dim}}$ is the number of dimensions of the Gaussian distribution. We implement the above bound in code instead of the non-specialized form of the KL divergence to speed up computations and simplify gradient computations.

### A.3.2 Few-Shot Learning Bound with Validation Data

In this section, we will assume that, in addition to the training data $S \sim P_{z|t}^m$, we have access to validation data $S_{\mathrm{va}} \sim P_{z|t}^n$ at meta-training time. We will show that a meta-learning generalization bound can still be obtained in this case. Notably, this will not require validation data at meta-testing time.

We begin by bounding the expected loss on evaluation data $S_{\mathrm{ev}} = \{S, S_{\mathrm{va}}\}$ after training on $S$. Note that for other meta-learning techniques, the training data $S$ is often excluded from the data used to update the meta-learner. Including it here helps relate the loss on $P_{z|t}$ to the loss on $S_{\mathrm{ev}}$ after adaptation with $S$ (see derivation below), and is necessary to achieve a guarantee on performance for the few-shot learning case. From Inequality (4), we set the arbitrary distribution $P_s$ to the marginal distribution $P_{S_{\mathrm{ev}}}$ over datasets of size $m + n$ and $f(\theta, s) := \widehat{L}(h_{A(\theta, S)}, S_{\mathrm{ev}})$. Note that with this marginal distribution, we have an equivalence of sampling given by

$$\mathop{\mathbb{E}}_{S_{\mathrm{ev}} \sim P_{S_{\mathrm{ev}}}} [\cdot] = \mathop{\mathbb{E}}_{t \sim P_t} \mathop{\mathbb{E}}_{S_{\mathrm{ev}} \sim P_{z|t}^{m+n}} [\cdot]. \tag{24}$$

The following inequality holds with high probability over a sampling of $\mathbf{S}_{\mathrm{ev}} = \{S_{\mathrm{ev},1}, S_{\mathrm{ev},2}, \dots, S_{\mathrm{ev},l}\} \sim P_{S_{\mathrm{ev}}}^l$:

$$\mathop{\mathbb{E}}_{S_{\mathrm{ev}} \sim P_{S_{\mathrm{ev}}}} \mathop{\mathbb{E}}_{\theta \sim P_\theta} \widehat{L}(h_{A(\theta, S)}, S_{\mathrm{ev}}) =$$

$$\mathop{\mathbb{E}}_{t \sim P_t} \mathop{\mathbb{E}}_{S_{\mathrm{ev}} \sim P_{z|t}^{m+n}} \mathop{\mathbb{E}}_{\theta \sim P_\theta} \widehat{L}(h_{A(\theta, S)}, S_{\mathrm{ev}}) \le \frac{1}{l} \sum_{i=1}^l \mathop{\mathbb{E}}_{\theta \sim P_\theta} \widehat{L}(h_{A(\theta, S_i)}, S_{\mathrm{ev},i}) + R_{\mathrm{PAC-B}}(P_\theta, P_{\theta,0}, \delta, l).$$

$$\tag{25}$$

In the next steps, we aim to isolate for a $\widehat{L}(h_{A(\theta, S)}, S)$ term so that we may still combine with Inequality (8) as we did in Section 4.2. We decompose the LHS of Inequality (25),

$$\frac{1}{m+n} \mathop{\mathbb{E}}_{t \sim P_t} \left[ m \mathop{\mathbb{E}}_{S \sim P_{z|t}^m} \mathop{\mathbb{E}}_{\theta \sim P_\theta} \widehat{L}(h_{A(\theta, S)}, S) + n \mathop{\mathbb{E}}_{S \sim P_{z|t}^m} \mathop{\mathbb{E}}_{S_{\mathrm{va}} \sim P_{z|t}^n} \mathop{\mathbb{E}}_{\theta \sim P_\theta} \widehat{L}(h_{A(\theta, S)}, S_{\mathrm{va}}) \right]. \tag{26}$$

Since the validation data $S_{\mathrm{va}}$ is sampled independently from $S$, the expected training loss on the validation data is the true expected loss over sample space $P_{z|t}$,

$$\mathop{\mathbb{E}}_{S \sim P_{z|t}^m} \mathop{\mathbb{E}}_{S_{\mathrm{va}} \sim P_{z|t}^n} \mathop{\mathbb{E}}_{\theta \sim P_\theta} \widehat{L}(h_{A(\theta, S)}, S_{\mathrm{va}}) = \mathop{\mathbb{E}}_{S \sim P_{z|t}^m} \mathop{\mathbb{E}}_{\theta \sim P_\theta} L(h_{A(\theta, S)}, P_{z|t}). \tag{27}$$

We plug Equality (27) into Equation (26), and then the decomposition in Equation (26) into Inequality (25). We can then isolate for the $\widehat{L}(h_{A(\theta,S)}, S)$ term,

$$\mathbb{E}_{t \sim P_t} \mathbb{E}_{S \sim P_{z|t}^m} \mathbb{E}_{\theta \sim P_\theta} \widehat{L}(h_{A(\theta,S)}, S) \leq \frac{m+n}{m} \left[ \frac{1}{l} \sum_{i=1}^{l} \mathbb{E}_{\theta \sim P_\theta} \widehat{L}(h_{A(\theta,S_i)}, S_{\text{ev},i}) + R_{\text{PAC}-\text{B}}(P_\theta, P_{\theta,0}, \delta, l) \right]$$
$$- \frac{n}{m} \mathbb{E}_{t \sim P_t} \mathbb{E}_{S \sim P_{z|t}^m} \mathbb{E}_{\theta \sim P_\theta} L(h_{A(\theta,S)}, P_{z|t}), \tag{28}$$

and plug into the LHS of Equation (8). By simplifying, we find that

$$\mathbb{E}_{t \sim P_t} \mathbb{E}_{S \sim P_{z|t}^m} \mathbb{E}_{\theta \sim P_\theta} L(h_{A(\theta,S)}, P_{z|t}) \leq \frac{1}{l} \sum_{i=1}^{l} \mathbb{E}_{\theta \sim P_\theta} \widehat{L}(h_{A(\theta,S_i)}, S_{\text{ev},i}) + R_{\text{PAC}-\text{B}}(P_\theta, P_{\theta,0}, \delta, l) + \frac{m\beta_{\text{US}}}{m+n}. \tag{29}$$

This resulting bound is very similar to the one in Inequality (6). We compute the loss term in the upper bound with evaluation data and as a result, the size of the uniform stability regularization term is reduced.

## A.4  Bounds on the Uniform Stability Constant

In this section, we present bounds from [29] on the uniform stability constant $\beta_{\text{US}}$ which are applicable to our settings. We first formalize the definitions of Lipschitz continuous ("Lipschitz" with constant $c_L$) and Lipschitz smoothness ("smooth" with constant $c_S$).

**Definition 3** ($c_L$-Lipschitz) *Function f is $c_L$-Lipschitz if $\forall \theta, \theta' \in \mathbb{R}^{n_\theta}, \forall z \in \mathcal{Z}$ the following holds:*

$$|f(\theta, z) - f(\theta', z)| \leq c_L \|\theta - \theta\|. \tag{30}$$

**Definition 4** ($c_S$-smooth) *Function f is $c_S$-smooth if $\forall \theta, \theta' \in \mathbb{R}^{n_\theta}, \forall z \in \mathcal{Z}$ the following holds:*

$$\|\nabla f(\theta, z) - \nabla f(\theta', z)\| \leq c_S \|\theta - \theta'\|. \tag{31}$$

Using a convex loss and stochastic gradient descent (SGD) allows us to directly bound the uniform stability constant $\beta_{\text{US}}$ [29]:

**Theorem 4** (Convex Loss SGD is Uniformly Stable [29]) *Assume that convex loss function $L$ is $c_S$-smooth and $c_L$-Lipschitz $\forall z \in \mathcal{Z}$. Suppose we run SGD on $S$ with step size $\alpha \leq \frac{2}{c_S}$ for $T$ steps. Then SGD satisfies $\beta_{\text{US}}$ uniform stability with*

$$\beta_{\text{US}} \leq \frac{2c_L^2}{m} T\alpha. \tag{32}$$

Note that the bounds on $\beta_{\text{US}}$ presented in [29] guarantee $\beta_{\text{US}}$ uniform stability in expectation for a randomized algorithm A. However, for deterministic algorithms, this reduces to $\beta_{\text{US}}$ uniform stability. Using the uniform stability in expectation definition introduces another expectation (over a draw of algorithm $A$) into the upper bound of the meta-learning generalization guarantee in Inequality (6). So as to not increase the computation required to estimate the upper bound, we let $A$ be deterministic. This is achieved either by fixing the order of the samples on which we perform gradient updates for SGD, or by using gradient descent (GD). Additionally, in the convex case, $T$ steps of GD satisfies the same bound on $\beta_{\text{US}}$ as $T$ steps of SGD; see Appendix A.8.1 for the proof. For non-convex losses, a bound on $\beta_{\text{US}}$ is still achieved when algorithm $A$ is SGD [29]:

**Theorem 5** (Non-Convex Loss SGD is Uniformly Stable [29]) *Let non-convex loss $L$ be $c_S$-smooth and $c_L$-Lipschitz $\forall z \in P_{z|t}$ and satisfy Assumption 1. Suppose we run $T$ steps of SGD with monotonically non-increasing step size $\alpha_t \leq \frac{c}{t}$. Then SGD satisfies $\beta_{\text{US}}$ uniform stability with*

$$\beta_{\text{US}} \leq \frac{1 + \frac{1}{c_S c}}{n-1} (2c_L^2 c)^{\frac{1}{c_S c+1}} T^{\frac{c_S c}{c_S c+1}} \tag{33}$$

Note that this bound does not hold when GD is used.

---

**Algorithm 2** PAC-BUS using Mini-Batches of Tasks

---
**Input**: Fixed prior distribution $\mathcal{N}_{\psi_0}$ over initializations
**Input**: $\beta_{\text{US}}$ uniformly stable Algorithm $A$
**Input**: Meta-training dataset $\mathbf{S} = \{S_1, S_2, \ldots, S_l\}$, learning rate $\gamma$
**Initialize**: $\psi \leftarrow \psi_0$
**Output**: Optimized $\psi^*$
$B(\psi, \theta_1', \theta_2' \ldots, \theta_k') := \frac{1}{l} \sum_{i=1}^{k} \widehat{L}(h_{\theta_i'}, S_i) + R_{\text{PAC-B}}(\mathcal{N}_\psi, \mathcal{N}_{\psi_0}, \delta, l) + \beta_{\text{US}}$
**while** not converged **do**
    Sample $\theta \sim \mathcal{N}_\psi$
    **for** $i = 1$ **to** $k$ **do**
        $j \sim \text{Uniform}\{1, 2, \ldots, l\}$
        $\theta_i' \leftarrow A(\theta, S_j)$
    **end for**
    $\psi \leftarrow \psi - \gamma \nabla_\psi B(\psi, \theta_1', \theta_2' \ldots, \theta_k')$
**end while**

---

## A.5 Algorithms

Before running the algorithms presented in this paper, we must deal with a few technical challenges that arise from our method's assumptions and terms which need to be computed. In this paragraph, we discuss the approach we take to deal with these challenges. For arbitrary networks, the softmax-activated cross entropy loss ($\text{CEL}_s$) is not bounded and would not satisfy Assumption 1. We thus constrain the network parameters to lie within a ball and scale the loss function such that all samples $z \in \mathcal{Z}$ achieve a loss within $[0, 1]$; see Appendix A.7 for details. However, the PAC-BUS framework works with distributions $P_\theta$ over initializations. One option is to let $P_\theta$ be a projected multivariate Gaussian distribution. This prevents the network's output from becoming arbitrarily large. However, the upper bound in Inequality (9) requires the KL-divergence between the prior and posterior distribution over initializations. This is difficult to calculate for projected multivariate Gaussian distributions and would require much more computation during gradient steps. Since the KL-divergence between projected Gaussians is less than that between Gaussians (due to the data processing inequality [21]), we can loosen the upper bound in (6) and (9) by computing the upper bound using the non-projected distributions (but using the projected Gaussians for the algorithm). After sampling a base learner's initialization, we re-scale the network such that its parameters lie within a ball of radius $r$. We also re-scale the base learner's parameters after each gradient step to guarantee that the loss stays within $[0, 1]$. Projection after gradient steps is not standard SGD, but we show that it maintains the same bound on $\beta_{\text{US}}$; see Section A.8.2 for details of the proof. Thus, we let algorithm $A$ be SGD with projections after each update and use Theorem 5 to bound $\beta_{\text{US}}$ for non-convex losses [29]. Additionally, we can upper bound the Lipschitz $c_L$ and smoothness $c_S$ constants for the network using the methods presented in [68]. After working through these technicalities, we can compute all terms in the upper bound.

### A.5.1 PAC-BUS using Mini-Batches of Tasks

We present the PAC-BUS algorithm modified for mini-batches of tasks to improve training times. For batches of size $k$, the algorithm is presented in 2.

### A.5.2 PAC-BUS(H)

In addition to providing algorithms which minimize the upper bound in Inequalities (6) and (9), we are also interested in a regularization scheme which re-weights the regularizer terms in these bounds. For larger scale and complex settings, it is challenging to provide a non-vacuous guarantee on performance, but weighting regularizer terms has been shown to be an effective training technique [77]. We calculate $\beta_{\text{US}}$ with a one-gradient-step version of Theorem 5. This Theorem requires the algorithm $A$ to be SGD, but we let $A$ be a single step of GD to improve training times. We also relax Assumption 1 Since the $\beta_{\text{US}}$ depends on both $c_L$ and $c_S$, we update estimates of them after each iteration by sampling multiple $\theta \sim P_\theta$, bound the $c_L$ and $c_S$ for those sets of parameters using Section 4 of [68], and then choose the maximum to compute $\beta_{\text{US}}$. This is in contrast to limiting the network parameters directly by bounding the output of the loss. Instead, the $\beta_{\text{US}}$ term in the upper

**Algorithm 3** PAC-BUS(H): Meta-learning heuristic based on PAC-BUS upper bound

---

**Input**: Fixed prior distribution $\mathcal{N}_{\psi_0}$ over initializations
**Input**: Meta-training dataset $\mathbf{S}$, learning rates $\alpha$ and $\gamma$
**Input**: Scale factors $\lambda_1, \lambda_2$ for regularization terms
**Initialize**: $\psi \leftarrow \psi_0$
**Output**: Optimized $\psi^*$
$B(\psi, c_L, c_S, \theta'_1, \theta'_2, \ldots, \theta'_l) := \frac{1}{l} \sum_{i=1}^{l} \widehat{L}(h_{\theta'_i}, S_i) + \lambda_1 R_{\text{PAC-B}}(\mathcal{N}_\psi, \mathcal{N}_{\psi_0}, \delta, l) + \lambda_2 \beta_{\text{US}}(c_L, c_S)$
Estimate $c_L$ and $c_S$ using $\mathcal{N}_{\psi_0}$
**while** not converged **do**
    Sample $\theta \sim \mathcal{N}_\psi$
    **for** $i = 1$ **to** $l$ **do**
        $\theta'_i \leftarrow \theta - \alpha \nabla_\theta \widehat{L}(h_\theta, S_i)$
    **end for**
    $\psi \leftarrow \psi - \gamma \nabla_\psi B(\psi, c_L, c_S, \theta'_1, \theta'_2, \ldots, \theta'_l)$
    Estimate $c_L$ and $c_S$ using $\mathcal{N}_\psi$
**end while**

---

bound and the scale factor will determine how much to restrict the network parameters. The resulting method is presented in Algorithm 3. In order to provide strong performance in practice, we tune $\lambda_1$ and $\lambda_2$.

### A.6 Sample Convergence Bound

After training is complete, we aim to compute the upper bound. However, this requires evaluating an expectation $\theta \sim P_\theta$, which may be intractable. Providing a valid PAC guarantee without needing to evaluate the expectation taken over $\theta \sim P_\theta$ requires the use of the sample convergence bound [37]. We have the following guarantee with probability $1 - \delta'$ over a random draw of $\{\theta_1, \theta_2 \ldots, \theta_N\} \sim P_\theta^N$ for any dataset $S$ [37],

$$D_{\text{KL}}\left( \sum_{j=1}^{N} L(h_{A(\theta_j, S)}, \cdot) \middle\| \mathbb{E}_{\theta \sim P_\theta} L(h_{A(\theta, S)}, \cdot) \right) \leq \frac{\log(\frac{2}{\delta'})}{N}. \tag{34}$$

We can invert this KL-style bound (i.e. a bound of the form $D_{\text{KL}}(p\|q^*) \leq c$) by solving the optimization problem, $q^* \leq D_{\text{KL}}^{-1}(q\|c) := \sup\{q \in [0,1] : D_{\text{KL}}(p\|q) \leq c\}$, as described in [24]. After the inversion is performed on Inequality (34), we use a union bound to combine the result with Inequality (6) and retain a guarantee with probability $1 - \delta - \delta'$ as in [24],

$$\mathcal{L}(P_\theta, P_t) \leq \frac{1}{l} \sum_{i=1}^{l} D_{\text{KL}}^{-1}\left( \sum_{j=1}^{N} L(h_{A(\theta_j, S_i)}, S_i) \middle\| \frac{\log(\frac{2}{\delta'})}{N} \right) + R_{\text{PAC-B}}(P_\theta, P_{\theta,0}, \delta, l) + \beta_{\text{US}}. \tag{35}$$

An analogous bound is achieved when combined with Inequality (9). Thus, after training, we evaluate Inequality (35) to provide the guarantee. Note that use of the sample convergence bound is a loosening step. However, in our experiments, the upper bound in Inequality (35) is less than 5% looser than unbiased estimates of Inequality (6). This can be reduced further at the expense of computation time (if we utilize a larger number of samples in the concentration inequality).

### A.7 Constraining Parameters and Scaling Losses

In order to maintain a guarantee, the PAC-Bayes upper bound in Theorem 2 requires a loss function bounded between 0 and 1. However, the losses we use are not bounded in general. Let $N_\theta$ be an arbitrary network parameterized by $\theta$ and $N_\theta(z)$ be the output of the network given sample $z \in \mathcal{Z}$. Consider arbitrary loss $f$, which maps the network's output to a real number. If $\|N_\theta(z)\| \leq r, \forall\, \theta \in \mathbb{R}^{n_\theta}, \forall\, z \in \mathcal{Z}$, then we can perform a linear scaling of $f$ to map it onto the interval $[0, 1]$. We define the minimum and maximum value achievable by loss function $f$ as follows

$$M_f := \max_{z \in \mathcal{Z},\, \theta \in \mathbb{R}^{n_\theta},\, \|N_\theta(z)\| \leq r} f(\theta, z) \tag{36}$$

$$m_f := \min_{z \in \mathcal{Z},\, \theta \in \mathbb{R}^{n_\theta},\, \|N_\theta(z)\| \leq r} f(\theta, z). \tag{37}$$

Now we can define a scaled function

$$f_S(\theta, z) := \frac{f(\theta, z) - m_f}{M_f - m_f} \tag{38}$$

such that $f_S(\theta, z) \in [0, 1]$. Note that the Lipschitz and smoothness constants of $f_S$ are also scaled by $\frac{1}{M_f - m_f}$. When we choose loss $\text{CEL}_s$, the $k$-class cross entropy loss with softmax activation, we have

$$M_{\text{CEL}_s} := \log\left(\frac{e^{-r} + (k-1)}{e^{-r}}\right), \quad m_{\text{CEL}_s} := \log\left(\frac{e^{r} + (k-1)}{e^{r}}\right). \tag{39}$$

However, we must restrict the parameters in such a way that satisfies $\|N_\theta(z)\| \le r$. For arbitrary networks structures, this is not straightforward, so we only analyze the case we use in this paper. Consider an $L$-layer network with ELU activation. Let parameters $\theta$ contain weights $\mathbf{W}_1, \ldots, \mathbf{W}_L$, and biases $b_1, \ldots, b_L$, and assume bounded input $\|z\| \le r_z, \forall\ \mathcal{Z}$.

$$\|N_\theta(z)\| = \|\text{ELU}\big(\mathbf{W}_L \text{ELU}(\mathbf{W}_{L-1}(\cdots) + b_{L-1}) + b_L\big)\| \tag{40}$$

$$\le \|\mathbf{W}_L\|_F(\|\mathbf{W}_{L-1}\|_F(\cdots) + \|b_{L-1}\|) + \|b_L\| \le r \tag{41}$$

We can satisfy $\|N_\mathbf{W}(z)\| \le r$ by restricting

$$\|\theta\|^2 = \sum_{i=1}^{L} \|\mathbf{W}_i\|_F^2 + \sum_{i=1}^{L} \|b_i\|^2 \le \left(\frac{r}{\max(1, r_z)}\right)^2. \tag{42}$$

Equation (42) implies Equation (41) by applying the inequality of arithmetic and geometric means. Thus, we ensure $\|\theta\| \le r/\max(1, r_z)$ by projecting the network parameters onto the ball of radius $\min(r, \frac{r}{r_z})$ after each gradient update.

## A.8 Uniform Stability Considerations

### A.8.1 Uniform Stability for Gradient Descent

In this section, we will prove that $T$ steps of GD has the same uniform stability constant as $T$ steps of SGD in the convex case. This will allow us to use GD when attempting to minimize a convex loss, Section 5.2. Let the gradient update rule $G$ be given by $G(\theta, z) = \theta - \alpha \nabla_\theta f(\theta, z)$ for *convex* loss function $f$, initialization $\theta \in \mathbb{R}^{n_\theta}$, sample $z \in \mathcal{Z}$, and positive learning rate $\alpha$. We define two key properties for gradient updates: expansiveness and boundedness [29].

**Definition 5** ($c_E$-expansive, Definition 2.3 in [29]) *Update rule $G$ is $c_E$-expansive if $\forall\ \theta, \theta' \in \mathbb{R}^{n_\theta}, \forall\ z \in \mathcal{Z}$ the following holds:*

$$\|G(\theta, z) - G(\theta', z)\| \le c_E \|\theta - \theta'\|. \tag{43}$$

**Definition 6** ($c_B$-bounded, Definition 2.4 in [29]) *Update rule $G$ is $c_B$-bounded if $\forall\ \theta \in \mathbb{R}^{n_\theta}, \forall\ z \in \mathcal{Z}$ the following holds:*

$$\|\theta - G(\theta, z)\| \le c_B. \tag{44}$$

Now, consider dataset $S \in \mathcal{Z}^m$ and define $\bar{f}(\theta, S) := \frac{1}{m} \sum_{i=1}^{m} f(\theta, z_i)$. We also define $\bar{G}(\theta, S) := \theta - \alpha \nabla_\theta \bar{f}(\theta, S) = \sum_{i=1}^{m} G(\theta, z_i)$. Assume that $G(\theta, z)$, is $c_E$-expansive and $c_B$-bounded $\forall\ z \in \mathcal{Z}$. We then bound the expansiveness of $\bar{G}(\theta, S)$,

$$\|\bar{G}(\theta, S) - \bar{G}(\theta', S)\| \le \frac{1}{m} \sum_{i=1}^{m} \|G(\theta, z_i) - G(\theta', z_i)\| \le \frac{1}{m} \sum_{i=1}^{m} c_E \|\theta - \theta'\| = c_E \|\theta - \theta'\|. \tag{45}$$

To compute the boundedness, consider

$$\|\theta - \bar{G}(\theta, S)\| \le \frac{1}{m} \sum_{i=1}^{m} \|\theta - G(\theta, z_i)\| \le \frac{1}{m} \sum_{i=1}^{m} c_B = c_B. \tag{46}$$

For a single gradient step on sample $z$, we see the same bounds on $c_E$ and $c_B$ when performing a single GD step on dataset $S$. Thus, if Lemmas 2.5, 3.3, and 3.7 in [29] are true for gradient updates $G$, they are also true for gradient updates $\bar{G}$. We can then run through the proof of Theorem 3.8 in [29] to show that it holds for $T$ steps of GD if it holds for $T$ steps of SGD.

Let $S \sim P_S$ be a dataset of size $m$ and $S'$ be an identical dataset with one element changed. We run $T$ steps of GD updates, $\bar{G}$, on each of $S$ and $S'$. This results in parameters $\theta_1, \ldots, \theta_T$ and $\theta_1', \ldots, \theta_T'$ respectively. Fix learning rate $\alpha \leq \frac{2}{c_S}$ and consider

$$\mathop{\mathbb{E}}_{S,S'} \|\theta_{t+1} - \theta_{t+1}'\| = \mathop{\mathbb{E}}_{S,S'} \|\bar{G}(\theta_t, S) - \bar{G}(\theta_t', S')\| \tag{47}$$

$$\leq \frac{1}{m} \sum_{j=1, i \neq j}^{m} \mathop{\mathbb{E}}_{S,S'} \|G(\theta_t, z_j) - G(\theta_t', z_j)\| + \frac{1}{m} \mathop{\mathbb{E}}_{S,S'} \|G(\theta_t, z_i) - G(\theta_t', z_i))\| \tag{48}$$

$$\leq \frac{m-1}{m} \mathop{\mathbb{E}}_{S,S'} \|\theta_t - \theta_t'\| + \frac{1}{m} \mathop{\mathbb{E}}_{S,S'} \|\theta_t - \theta_t'\| + \frac{2\alpha c_L}{m} = \mathop{\mathbb{E}}_{S,S'} \|\theta_t - \theta_t'\| + \frac{2\alpha c_L}{m} \tag{49}$$

The steps above follow from Lemmas 2.5, 3.3, and 3.7 in [29] and the linearity of expectation. The rest of the proof follows naturally and results in a uniformly stable constant $\beta_{\mathrm{US}} \leq \frac{2c_L^2}{m} T \alpha$ for $T$ steps of GD. Thus, we have the following result.

**Corollary 1** *Assume that loss convex function $f$ is $c_S$-smooth and $c_L$ Lipschitz $\forall\, z \in \mathcal{Z}$. Suppose $T$ steps of SGD on $S$ satisfies $\beta_{\mathrm{US}}$ uniform stability. This implies that $T$ steps of GD on $S$ satisfies $\beta_{\mathrm{US}}$ uniform stability.*

### A.8.2 Uniform Stability Under Projections

Projecting parameters onto a ball after gradient updates does not constitute standard SGD nor GD, so we analyze the stability constant after $T$ steps of $G_P(\theta, z) = \mathrm{Proj}[\theta - \alpha \nabla_\theta f(\theta, z)]$. Assume $\|z\| \leq r_z, \forall\, z \in \mathcal{Z}$. The function Proj scales parameters to satisfy $\|\theta\| \leq \max(r, \frac{r}{r_z})$ if it is not already satisfied. See Appendix A.7 for an explanation of this restriction.

As in Appendix A.8.1, we compute bounds on the expansiveness and boundedness of $G_P$. Suppose $\theta$ is a vector containing all weights of an $L$-layer network. Network hyper-parameters such as learning rate and activation parameters do not need to be projected, so they will not be included. Assume that $\theta, \theta'$ already satisfy $\|N_\theta(z)\| \leq r, \forall\, z \in \mathcal{Z}$. Consider

$$\|G_P(\theta, z) - G_P(\theta', z)\| = \|\mathrm{Proj}(G(\theta, z)) - \mathrm{Proj}(G(\theta', z))\| \leq \|G(\theta, z) - G(\theta', z)\| \leq c_E \|\theta - \theta'\|. \tag{50}$$

Note that any required scaling is equivalent to orthogonal projection of the parameters onto a euclidean norm ball of radius $r$ in $R^d$, where $d$ is the number of parameters in the network. Thus, the first inequality follows from the fact that orthogonal projections onto closed convex sets satisfy the contractive property [63]. Next, consider

$$\|\theta - G_P(\theta, z)\| = \|\mathrm{Proj}(\theta) - \mathrm{Proj}(G_P(\theta, z))\| \leq \|\theta - G(\theta, z)\| \leq c_B. \tag{51}$$

The equality follows from the assumption that $\theta$ already satisfies the norm constraint. As above, the first inequality follows from the fact that the Proj function satisfies the contractive property [63].

With these bounds, gradient update $G_P$ satisfies Lemmas 2.5, 3.3, and 3.7 from [29] if $G$ does. Note that an analogous procedure can be used to show that scaling after a GD update, $\bar{G}_P$, also satisfies these Lemmas. When function $f$ or $\bar{f}$ is convex, the proof of Theorem 3.8 in [29] applies, and shows that using gradient updates $G_P$ or $\bar{G}_P$ achieve the same bound on the uniform stability constant $\beta_{\mathrm{US}}$. Thus, when $f$ is convex, we may use $G_P$ or $\bar{G}_P$ to compute updates and maintain the guarantee presented in Theorem 1. Suppose now that $f$ is not convex. Using Lemmas 2.5, 3.3, 3.7, and 3.11 from [29], the proof of Theorem 3.12 in [29] follows naturally to achieve a bound on SGD using projected gradient updates $G_P$ when $f$ is not convex.

### A.9 Lipschitz and Smoothness Constant Calculation

Recall Definitions 3 and 4 for a function which is $c_L$-Lipschitz and $c_S$-smooth from Appendix A.4. We define the softmax activation function.

**Definition 7** (Softmax Function) $s : \mathbb{R}^k \to \mathbb{P}^k$

$$s(u)_i = \frac{e^{u_i}}{\sum_{j=1}^{k} e^{u_j}}, \ \forall\, i. \tag{52}$$

Where every element in $\mathbb{P}^k$ is a probability distribution in $k$ dimensions (i.e. if $v \in \mathbb{P}^k$, then $\sum_{i=1}^k v_i = 1$ and $v_i \geq 0 \ \forall \ i$). Since the stability constant $\beta_{\text{US}}$ depends directly on the Lipschitz constant of the loss function, and $\beta_{\text{US}}$ appears in the regularizer of the final bound, we will be as tight as possible when bounding the Lipschitz constant to keep the generalization as tight as possible. Section 6.2 of [75] describes an approach for bounding the Lipschitz constant for the 2-class, sigmoid activated, cross entropy loss. We are interested in the k-class case with softmax activation, and also aim to bound the smoothness constant. We begin with a similar analysis to the one described in [75].

Given unit-length column vector $z \in \mathbb{R}^d$ and row vector $y \in \mathbb{P}^k$, with weight matrix $\mathbf{W} \in \mathbb{R}^{d \times k}$ (representing a single-layer network), the loss function is given by:

$$\text{CEL}_s(\mathbf{W}) = -\sum_{i=1}^k y_i \log(s(z^T \mathbf{W})_i). \tag{53}$$

Note that while $y$ is any probability distribution, in practice, $y$ will be an indicator vector, describing the correct label with a 1 in the index of the correct class and 0 elsewhere. However, the analysis that follows does not depend on this assumption.

We will take the Hessian of this loss to determine convexity and the Lipschitz constant. However, since the weights are given by a matrix, the Hessian would be a 4-tensor. To simplify the analysis, we will define

$$\mathbf{w} = \begin{bmatrix} \mathbf{W}_{:,1} \\ \mathbf{W}_{:,2} \\ \vdots \\ \mathbf{W}_{:,k} \end{bmatrix}. \tag{54}$$

Where $\mathbf{W}_{:,i}$ is the $i^{th}$ column of $\mathbf{W}$ such that $\mathbf{w} \in \mathbb{R}^{dk}$. We also let

$$\mathbf{z}(i)^T = \begin{bmatrix} \bar{0} & \dots & \bar{0} & z^T & \bar{0} & \dots & \bar{0} \end{bmatrix} \tag{55}$$

such that $z$ is placed in the $i^{th}$ group of $d$ elements and $\bar{0}$ is a row vector of $d$ zeros. Vector $\mathbf{z}(i) \in \mathbb{R}^{dk}$ since there are $k$ groups. With these definitions, we write the softmax activated network defined by $\mathbf{W}$ with input $z$:

$$s(z^T \mathbf{W})_i = \frac{e^{\mathbf{z}(i)^T \mathbf{w}}}{\sum_{j=1}^k e^{\mathbf{z}(j)^T \mathbf{w}}}. \tag{56}$$

We can simplify this by plugging in for the definition of $s$:

$$\text{CEL}_s(\mathbf{w}) := \text{CEL}_s(\mathbf{W}) = -\sum_{i=1}^k y_i \left[ \mathbf{z}(i)^T \mathbf{w} - \log\left( \sum_{j=1}^k e^{\mathbf{z}(j)^T \mathbf{w}} \right) \right] \tag{57}$$

$$= -\sum_{i=1}^k y_i \mathbf{z}(i)^T \mathbf{w} + \log\left( \sum_{i=1}^k e^{\mathbf{z}(i)^T \mathbf{w}} \right). \tag{58}$$

These are equivalent because $\sum_{i=1}^k y_i = 1$. For readability, we let $p_i := s(z^T \mathbf{W})_i$. With these preliminaries the Hessian will be a 2-tensor and the $\nabla_{\mathbf{w}}^3$ term will be a 3-tensor. We compute the gradient and Hessian and $\nabla_{\mathbf{w}}^3$ term:

$$\nabla_{\mathbf{w}} \text{CEL}_s(\mathbf{w}) = -\sum_{i=1}^k y_i \mathbf{z}(i) + \sum_{i=1}^k \mathbf{z}(i) p_i \tag{59}$$

$$\nabla_{\mathbf{w}}^2 \text{CEL}_s(\mathbf{w}) = \sum_{i=1}^k \mathbf{z}(i) \mathbf{z}(i)^T p_i - \left( \sum_{i=1}^k \mathbf{z}(i) p_i \right) \left( \sum_{j=1}^k \mathbf{z}(j)^T p_j \right). \tag{60}$$

We write $\nabla_{\mathbf{w}}^3 \text{CEL}_s(\mathbf{w})$ termwise to simplify notation:

$$\nabla_{\mathbf{w}}^3 \text{CEL}_s(\mathbf{w}) = \begin{cases} (p_i - 3p_i^2 + 2p_i^3) z \otimes z^T \otimes z^\perp & i = j = l \\ (-p_i p_l + 2p_i^2 p_l) z \otimes z^T \otimes z^\perp & i = j \neq l \\ (-p_j p_i + 2p_j^2 p_i) z \otimes z^T \otimes z^\perp & j = l \neq i \\ (-p_l p_j + 2p_l^2 p_j) z \otimes z^T \otimes z^\perp & l = i \neq j \\ (2p_i p_j p_l) z \otimes z^T \otimes z^\perp & i \neq j \neq l \end{cases} \tag{61}$$

Where $\otimes$ is the tensor product and $z \otimes z^T \otimes z^\perp \in \mathbb{R}^{d \times d \times d}$ is a 3-tensor with the abuse of notation: $z \in \mathbb{R}^{d \times 1 \times 1}$, $z^T \in \mathbb{R}^{1 \times d \times 1}$, and $z^\perp \in \mathbb{R}^{1 \times 1 \times d}$. Thus $\nabla_\mathbf{w}^3 \text{CEL}_s(\mathbf{w}) \in \mathbb{R}^{dk \times dk \times dk}$.

For twice-differentiable functions, the Lipschitz constant is given by the greatest eigenvalue of the Hessian. Correspondingly, the smoothness constant is given by the greatest eigenvalue of the $\nabla_\mathbf{w}^3$ term for thrice-differentiable functions. Thus, we aim to bound the largest value that the Rayleigh quotient can take for any unit-length vector $x$. For the Hessian:

$$x^T \nabla_\mathbf{w}^2 \text{CEL}_s(\mathbf{w})x \leq |x^T \nabla_\mathbf{w}^2 \text{CEL}_s(\mathbf{w})x| = \|x^T \nabla_\mathbf{w}^2 \text{CEL}_s(\mathbf{w})x\|_F \tag{62}$$

$$\leq \|x\|^2 \|\nabla_\mathbf{w}^2 \text{CEL}_s(\mathbf{w})\|_F = \|\nabla_\mathbf{w}^2 \text{CEL}_s(\mathbf{w})\|_F \tag{63}$$

$$= \sqrt{\sum_{i=1}^k \|zz^T\|_F (p_i - p_i^2)^2 + \sum_{i=1}^k \sum_{j=1,j\neq i}^k \|zz^T\|_F (p_i p_j)^2} \tag{64}$$

$$= \sqrt{\sum_{i=1}^k (p_i - p_i^2)^2 + \sum_{i=1}^k \sum_{j=1,j\neq i}^k (p_i p_j)^2}. \tag{65}$$

The Frobenius norm is maximized when $p_i = \frac{1}{k}$ for $k > 1$:

$$\|\nabla_\mathbf{w}^2 \text{CEL}_s(\mathbf{w})\|_F \leq \sqrt{k\left(\frac{1}{k} - \frac{1}{k^2}\right)^2 + k(k-1)\left(\frac{1}{k^2}\right)^2} \tag{66}$$

$$= \frac{\sqrt{k-1}}{k}. \tag{67}$$

Thus, for $\text{CEL}_s(\mathbf{w})$, the Lipschitz constant, $c_L \leq \frac{\sqrt{k-1}}{k}$ when $k > 1$. We can also show that the Rayleigh quotient is lower bounded by 0 by following analogous steps in [75] (these steps are omitted from this appendix), and thus $\text{CEL}_s(\mathbf{w})$ is convex. Next, we examine the Rayleigh quotient of the $\nabla_\mathbf{w}^3 \text{CEL}_s(\mathbf{w})$. Analogous to the procedure for the Hessian, we make use of a 3-tensor analog of the Frobenius norm: $\|M\|_{3,F} := \sqrt{\sum_{i=1}^k \sum_{j=1}^k \sum_{l=1}^k M(i,j,l)^2}$. Thus we have the following inequality

$$x^T \otimes [x^\perp \otimes \nabla_\mathbf{w}^3 \text{CEL}_s(\mathbf{w})] \otimes x \leq \|\nabla_\mathbf{w}^3 \text{CEL}_s(\mathbf{w})\|_{3,F}. \tag{68}$$

Since $\|z \otimes z^T \otimes z^\perp\|_{3,F} = 1$, we can write this as

$$\|\nabla_\mathbf{w}^3 \text{CEL}_s(\mathbf{w})\|_{3,F} \leq \left( \begin{array}{l} \sum_{i=1}^k (p_i - 3p_i^2 + 2p_i^3)^2 + \sum_{i=1}^k \sum_{j=1,j\neq i}^k (-p_i p_l + 2p_i^2 p_l)^2 \\ + \sum_{j=1}^k \sum_{l=1,l\neq j}^k (-p_j p_i + 2p_j^2 p_i)^2 + \sum_{l=1}^k \sum_{i=1,i\neq l}^k (-p_l p_j + 2p_l^2 p_j)^2 \\ + \sum_{i=1}^k \sum_{j=1,j\neq i}^k \sum_{l=1,l\neq j}^k (2p_i p_j p_l)^2. \end{array} \right)^{1/2} \tag{69}$$

This is maximized when $p_i = \frac{1}{k}$ for $k > 2$, which was verified with the symbolic integrator Mathematica [76]. Simplifying results in:

$$\|\nabla_\mathbf{w}^3 \text{CEL}_s(\mathbf{w})\|_{3,F} \leq \sqrt{\frac{(k-1)(k-2)}{k^3}}. \tag{70}$$

Thus for $\text{CEL}_s(\mathbf{w})$, the smoothness constant, $c_S \leq \sqrt{\frac{(k-1)(k-2)}{k^3}}$ when $k > 2$. When $k = 2$, $p_1, p_2 = \frac{1}{2} \pm \frac{\sqrt{3}}{6}$ and $c_S \leq \sqrt{\frac{2}{27}}$.

### A.10  Study on Base-learning Learning Rate and Number of Update Steps

In this section we present additional results on the performance of the algorithms with different iterations and learning rates using the same example setup as in Section 5.1. Note that we have

not used the sample convergence bound (see Appendix A.6) and present results for a single sample $\theta \sim P_\theta$. The true values of the upper bounds for MLAP-M [5], MR-MAML [77], and PAC-BUS (our method) are unlikely to change by more than 5% as the sample complexity bound does not loosen the guarantee very much. We present these results to provide a qualitative sense of the guarantees and their trends for varying base-learning learning rates and number of update steps.

Below we present test losses for MAML [25] (as a baseline), MLAP-M [5], MR-MAML [77], and PAC-BUS for base-learning rates ($\mathrm{lr}_b$) of $0.01$ to $10$ using $\{1, 3, 10\}$ adaptation steps.

| MAML Test Loss, $\mathrm{lr}_b =$ | 0.01 | 0.03 | 0.1 | 0.3 | 1 | 3 | 10 |
|---|---|---|---|---|---|---|---|
| Adaptation steps = 1 | 0.184±0.007 | 0.184±0.008 | 0.168±0.006 | 0.152±0.004 | 0.120±0.001 | 0.114±0.001 | 0.133±0.007 |
| Adaptation steps = 3 | 0.177±0.006 | 0.179±0.002 | 0.149±0.002 | 0.126±0.001 | 0.115±0.001 | 0.106±0.001 | 0.123±0.004 |
| Adaptation steps = 10 | 0.179±0.004 | 0.155±0.002 | 0.128±0.001 | 0.124±0.001 | 0.113±0.001 | 0.104±0.002 | 0.129±0.008 |

| MLAP-M Test Loss, $\mathrm{lr}_b =$ | 0.01 | 0.03 | 0.1 | 0.3 | 1 | 3 | 10 |
|---|---|---|---|---|---|---|---|
| Adaptation steps = 1 | 0.181±0.010 | 0.175±0.014 | 0.150±0.006 | 0.129±0.009 | 0.083±0.001 | 0.065±0.003 | 0.220±0.044 |
| Adaptation steps = 3 | 0.178±0.006 | 0.159±0.007 | 0.102±0.005 | 0.081±0.003 | 0.064±0.001 | 0.050±0.004 | 0.379±0.021 |
| Adaptation steps = 10 | 0.161±0.005 | 0.115±0.002 | 0.078±0.004 | 0.063±0.002 | 0.050±0.001 | 0.045±0.002 | 0.919±0.036 |

| MR-MAML Test Loss, $\mathrm{lr}_b =$ | 0.01 | 0.03 | 0.1 | 0.3 | 1 | 3 | 10 |
|---|---|---|---|---|---|---|---|
| Adaptation steps = 1 | 0.171±0.003 | 0.169±0.003 | 0.163±0.003 | 0.146±0.002 | 0.127±0.001 | 0.128±0.000 | 0.178±0.008 |
| Adaptation steps = 3 | 0.170±0.002 | 0.166±0.001 | 0.146±0.002 | 0.128±0.001 | 0.123±0.001 | 0.118±0.001 | 0.163±0.022 |
| Adaptation steps = 10 | 0.165±0.002 | 0.152±0.002 | 0.129±0.001 | 0.126±0.001 | 0.118±0.001 | 0.115±0.001 | 0.139±0.009 |

| PAC-BUS Test Loss, $\mathrm{lr}_b =$ | 0.01 | 0.03 | 0.1 | 0.3 | 1 | 3 | 10 |
|---|---|---|---|---|---|---|---|
| Adaptation steps = 1 | 0.176±0.002 | 0.171±0.003 | 0.160±0.001 | 0.145±0.002 | 0.127±0.001 | 0.129±0.002 | 0.164±0.019 |
| Adaptation steps = 3 | 0.170±0.002 | 0.165±0.002 | 0.145±0.001 | 0.129±0.002 | 0.123±0.001 | 0.120±0.001 | 0.144±0.014 |
| Adaptation steps = 10 | 0.163±0.001 | 0.150±0.001 | 0.130±0.001 | 0.126±0.002 | 0.119±0.002 | 0.115±0.002 | 0.130±0.004 |

Next, we present the computed bounds for MLAP-M [5], MR-MAML [77], and PAC-BUS for the same set of hyper-parameters.

| MLAP-M Bound, $\mathrm{lr}_b =$ | 0.01 | 0.03 | 0.1 | 0.3 | 1 | 3 | 10 |
|---|---|---|---|---|---|---|---|
| Adaptation steps = 1 | **1.003±0.000** | 1.015±0.001 | 1.223±0.020 | 1.946±0.043 | 3.113±0.154 | 5.435±0.220 | 21.874±0.420 |
| Adaptation steps = 3 | 1.008±0.000 | 1.087±0.027 | 1.864±0.062 | 3.072±0.157 | 4.147±0.095 | 6.760±0.233 | 28.356±1.826 |
| Adaptation steps = 10 | 1.050±0.006 | 1.535±0.044 | 2.574±0.064 | 4.009±0.107 | 5.98±0.057 | 10.119±0.087 | 47.971±1.346 |

| MR-MAML Bound, $\mathrm{lr}_b =$ | 0.01 | 0.03 | 0.1 | 0.3 | 1 | 3 | 10 |
|---|---|---|---|---|---|---|---|
| Adaptation steps = 1 | 0.344±0.002 | 0.343±0.002 | 0.335±0.002 | 0.320±0.001 | 0.300±0.000 | 0.303±0.001 | 0.351±0.006 |
| Adaptation steps = 3 | 0.344±0.002 | 0.340±0.002 | 0.320±0.002 | 0.302±0.002 | 0.296±0.001 | **0.292±0.001** | 0.335±0.018 |
| Adaptation steps = 10 | 0.339±0.001 | 0.324±0.002 | 0.303±0.000 | 4.752±0.808 | 5.330±0.187 | 6.316±0.639 | 9.134±1.448 |

| PAC-BUS Bound, $\mathrm{lr}_b =$ | 0.01 | 0.03 | 0.1 | 0.3 | 1 | 3 | 10 |
|---|---|---|---|---|---|---|---|
| Adaptation steps = 1 | 0.216±0.002 | 0.216±0.002 | 0.204±0.002 | 0.188±0.002 | **0.169±0.000** | 0.171±0.001 | 0.207±0.021 |
| Adaptation steps = 3 | 0.252±0.001 | 0.247±0.002 | 0.228±0.002 | 0.211±0.001 | 0.204±0.001 | 0.200±0.002 | 0.228±0.017 |
| Adaptation steps = 10 | 0.383±0.002 | 0.372±0.001 | 0.350±0.001 | 1.160±0.093 | 1.288±0.056 | 1.650±0.055 | 2.221±0.256 |

These results show the dependence that the PAC-BUS upper bound (specifically the uniform stability regularizer term $\beta_{\mathrm{US}}$) has on the learning rate and number of base-learning update steps whereas the bound for MR-MAML does not suffer with increasing base-learning steps or learning rate. However, once the learning rate and number of adaptation steps are too large, all bounds worsen significantly. The tightest guarantee obtained using PAC-BUS is significantly stronger than those for any tuning of MR-MAML and MLAP-M. We bold the tightest guarantee achieved in the tables above to highlight this.

## A.11  Additional Experimental Details

In this section, we report information about the data used, the procedure for prior, train, and test splits, as well as other experimental details. Code capable of reproducing the results in this paper is publicly available at https://github.com/irom-lab/PAC-BUS. All results provided in this

paper were computed on an Amazon Web Services (AWS) p2 instances. Tuning and intermediate results were computed on a desktop computer with a 12-core Intel i7-8700k CPU and an NVIDIA Titan Xp GPU. In addition, we made use of several existing software assets: SciKit-learn [50] (BSD license), PyTorch [49] (BSD license), CVXPY [23, 2] (Apache License, Version 2.0), MOSEK [44] (software was used with a personal academic license, see `https://www.mosek.com/products/license-agreement` for more details), learn2learn [7] (MIT License), and h5py [20] (Python license, see `https://docs.h5py.org/en/stable/licenses.html` for more details).

### A.11.1 Circle Class

We randomly sample points from the unit ball $B^2(0, 1)$ and classify them as $(+)$ or $(-)$ according to whether or not the points are outside the ball $B^2(c_t, r_t)$. For the tasks which are used to train a prior, we sample $c_t$ from $[0.1, 0.5]$ and $r_t$ from $[0.1, 1 - \|c_t\|]$. For the meta-training and meta-testing tasks, we sample $c_t$ from $[0.1, 0.4]$ and $r_t$ from $[0.1, 1 - \|c_t\|]$.

For all methods, we train the prior on 500 tasks, train the network on 10000 tasks, and test on 1000 tasks. We report the meta-test loss and a guarantee on the loss if applicable. A single task is a 2-class 10-sample (i.e. there are 10 samples given in total for training, not 10 samples from each class) learning problem. The evaluation dataset $S_{ev}$ consists of a dataset $S$ of 10 base-learner training samples and a dataset $S_{va}$ of 250 validation samples. For PAC-BUS, we searched for the meta-learning rate in $[1e-4, 1]$, the base-learning rate in $[0.01, 10]$, and the number of base-learning update steps in $[1, 10]$. The resulting parameters for the 10-shot learning problems are: meta-learning rate $1e-3$, base-learning rate $0.05$, and 1 base-learning update step. Note that in this example and the *Mini-wiki* example, we select the number of base-learning steps such that the upper bound is minimized. A lower loss may have been achievable with more base-learning update steps, but we aim to produce the tightest bound possible. Training for each method took less than 1 hour on the AWS p2 instance and computing the sample convergence upper bound took approximately 3 days when applicable.

### A.11.2 Mini-wiki

In Table 4, we present additional results – the percentage of correctly classified sentences on test tasks (after the base learner's adaptation step). Note that we present these results with the same posterior as was used to generate the results in Table 2.

Table 4: Meta-test accuracy as a percentage for MAML, FLI-Batch, MR-MAML, and PAC-BUS. We report the mean and standard deviation after 5 trials.

| 4-WAY *Mini-Wiki* | 1-SHOT ↑ | 3-SHOT ↑ | 5-SHOT ↑ |
|---|---|---|---|
| MAML [25] | **60.2 ± 0.9** | 68.3 ± 0.7 | **71.9 ± 0.6** |
| FLI-BATCH [34] | 46.0 ± 5.9 | 48.7 ± 4.9 | 54.5 ± 2.4 |
| MR-MAML [77] | 59.9 ± 0.8 | **68.4 ± 0.7** | 71.8 ± 0.7 |
| PAC-BUS (OURS) | 59.9 ± 0.8 | 68.1 ± 0.7 | 71.2 ± 0.7 |

We use the *Mini-wiki* dataset from [34], which consists of 813 classes each with at least 1000 example sentences from that class's corresponding Wikipedia article. The dataset was derived from the Wiki3029 dataset presented in [9], which was created from a public domain (CC0 license) Wikipedia dump. Although the Wikipedia dump is open source, it is possible that content which is copyrighted was used since the datasets are large and it is difficult to moderate all content on the website. In addition, it is possible that the dataset has some offensive content such as derogatory terms or curse words. However, since these are in the context Wikipedia articles, the authors trust that the original article was not written maliciously, but for the purposes of education. We use the first 62 classes of *Mini-wiki* for training the prior, the next 625 for the meta-training, and the last 126 for meta-testing. Before creating learning tasks, we remove all sentences with fewer than 120 characters.

For all methods, we train the prior on 100 tasks, train the network on 1000 tasks, and test on 200 tasks. We report the meta-test score, the meta-test loss, and a guarantee on the loss if applicable. A single task is a 4-class $\{1, 3, 5\}$-shot learning problem. The evaluation dataset $S_{ev}$ consists of a dataset $S$ of $\{1, 3, 5\}$ base-learner training samples and a dataset $S_{va}$ of $\{250, 250, 250\}$ validation samples respectively. For PAC-BUS, we search for the meta-learning rate in $[0.01, 1]$ the base-learning rate in $[1e-3, 100]$, and the number of base-learning update steps in $[1, 50]$. The resulting parameters for the

$\{1, 3, 5\}$-shot learning problems are: meta-learning rate $\{0.1, 0.1, 0.1\}$, base-learning rate $\{2.5, 5, 5\}$, and $\{2, 4, 5\}$ base-learning update steps respectively. Training for each method took less than 1 hour on the AWS p2 instance and computing the sample convergence upper bound took approximately 2 days when applicable.

### A.11.3 Omniglot

We use the *Omniglot* dataset from [35], which consists of 1623 characters each with 20 examples. The dataset was collected using Amazon's Mechanical Turk (AMT) and is available on GitHub with an MIT license. This dataset was collected voluntarily by AMT workers. Since the dataset is small enough, it can be checked visually for personally-identifiable information. We use the first 1200 characters for meta-training and the remaining 423 for meta-testing. The image resolution is reduced to $28 \times 28$. In the non-mutually exclusive setting, the 1200 training characters are randomly partitioned into 20 equal-sized groups which are assigned a fixed class label from 1 to 20. Note that this is distinct from the method described in [77] where the data is partitioned into 60 disjoint sets. Both experimental setups cause memorization, but the setup used in [77] causes more severe memorization than ours. This is why our implementation of MAML performs better than the results for MAML reported in [77]. However, our implementation of MR-MAML(W) method performs similarly to what is reported in [77].

For all methods, we trained on 100000 batches of 16 tasks and report the meta-test score on 8000 test tasks. We also used 5 base-learning update steps for all methods. A single task is a 20-way $\{1, 5\}$-shot learning problem. The evaluation dataset $S_{\mathrm{ev}}$ consists of a dataset $S$ of $\{1, 5\}$ base-learner training samples and a dataset $S_{\mathrm{va}}$ of $\{4, 5\}$ validation samples respectively. For PAC-BUS(H), we searched for the regularization scales $\lambda_1$ and $\lambda_2$ in $[1e{-}7, 1]$ and $[1e{-}4, 1e4]$ respectively. Additionally, the meta-learning rate was selected from $[5e{-}4, 0.1]$, and the base-learning rate was selected from $[0.01, 10]$. The resulting parameters for the $\{1, 5\}$-shot learning problems are: $\lambda_1 = \{1e{-}3, 1e{-}4\}$, $\lambda_2 = \{10, 10\}$, meta-learning rate $\{1e{-}3, 1e{-}3\}$, and base-learning rate $\{0.5, 0.5\}$ respectively. Training for each method took approximately 3 days on the AWS p2 instance.