# OpenReview forum: "Generalization Bounds for Meta-Learning via PAC-Bayes and Uniform Stability"
_NeurIPS.cc/2021/Conference — NeurIPS 2021 Poster_

### Official Review · Reviewer_mTwB · 2021-07-13

**Rating:** 7
**Confidence:** 5

**Summary:**

This submission is about generalization bounds for gradient-based meta-learning algorithms. It is motivated by the observations that meta-learning algorithms have demonstrated promising empirical results in a variety of tasks, while our theoretical understanding of these algorithms is lagging behind and in particular the existing generalization bounds for meta-learning are challenging to compute or give vacuous values even in simple settings.

I like how this work reasons about the challenges in generalization for meta-learning. The reasoning is by separately considering generalization at the base level and generalization at the meta level, and how the two levels must be coupled. The work proposes to rely on uniform stability to reason about generalization at the base level, and to use PAC-Bayes bounds to reason about generalization at the meta level. The claimed main contribution is that this approach leads to a novel generalization bound (Theorem 3) for gradient-based meta-learning, which is then leveraged to develop a learning method (PAC-BUS) by using the bound as optimization objective, and the method is evaluated in two meta-learning problems. It is not too clear to me how the two levels are coupled to learn the inductive biases that can be used for future tasks, but I suspect that the PAC-Bayes prior must play a role in connecting the two levels.

The argument behind the proof of the main result (Theorem 3) appears to consist of three blocks: (1) writing the distribution that generates data sets of a given size $m$ as a product distribution between the random tasks and the random samples for given tasks; (2) using the classical PAC-Bayes bound of McAllester for bounding the expected loss (expectation over data sets and over parameters) in terms of its empirical counterpart (that sums over the training tasks) plus a KL regularization term; and (3) using Uniform Stability for bounding the expected loss on a given task in terms of its empirical counterpart and stability. However, I missed an outline of the proof or indication of where the reader can see the full details of the proof.

**Limitations And Societal Impact:**

Sure.

**Main Review:**

ORIGINALITY

Generalization bounds for meta-learning have been proposed before. There are stability-based approaches (works of Maurer) and approaches that are based on PAC-Bayes bounds (Amit and Meir). However, the combination of these two approaches seems to be a novel idea.

SIGNIFICANCE

Assuming that the main result is correct, this is an interesting result conceptually and potentially also with respect to the improved tightness of the numerical bound values that it may lead to in practice. However, I would like to flag that the details of the proof are not provided, nor is the proof outlined at least. I would like to request the author(s) to include in their response an outline of the proof of Theorem 3, and in case this paper is accepted, the author(s) should include such outlined proof in the paper and an additional appendix giving the full details of the proof.

CLARITY / QUALITY / READABILITY

The paper is relatively well-written. The introduction is very nice and informative. However, I have some concerns regarding Section 2 on problem formulation, where the mathematical language is declared. It seems to me that the choices of notation affect the clarity/readability of the paper in a non-trivial way. I have made an effort to point out specific difficulties that I encountered and to suggest possible fixes. Many other comments of editorial nature are shared next.

EDITORIAL FEEDBACK

Title: Would the authors consider the title "Meta-Learning Bounds via PAC-Bayes and Uniform Stability" ?
(The acronym PAC-BUS can be introduced and used in the paper.)

L.8-9: "the result of this approach is a novel PAC bound [...]" (i.e. insert "of this approach" and remove the hyphen)

L.19: Replace "one" with "a learner"

L.23: "base learner" (no hyphen, I think)

L.36: Replace "realistic problems" with "benchmark problems" and also delete "ImageNet-scale"

L.43: "meta-learner" (twice)

L.48: Together with [11] it would be good to cite Bousquet et al. (2020) "Sharper bounds for uniformly stable algorithms"

L.50-51: I think the definition of "uniform stability" of Bousquet & Eliseeff [11] is in the sup norm (hence it makes sense that they called it "uniform"), while the definition of Hardt et al. [22] is in expectation (the L_1 norm of the difference) only. Since these two definitions do not coincide, this passage is potentially confusing/misleading without further clarification. Perhaps say that [22] demonstrated that limiting the number of training epochs [...] leads to stability in expectation.

L.52: delete "uniformly"
L.54: delete "uniform"

Possibly the work of Shalev-Shwartz and collaborators ("Learnability, stability and uniform convergence" and "Learnability and stability in the general learning setting") and perhaps work of Poggio and collaborators ("On learnability, complexity and stability") could be cited to support the connection between stability and learnability, on which you rely at the base level?

L.60-61: Replace "one has seen" with "that have been seen"

L.68-69: "We refer to the resulting approach as PAC-BUS since it combines PAC-Bayes and Uniform Stability to derive generalization guarantees for meta-learning."

Section 2: The math notation choice has an impact on the clarity/readability of the submission. I suggest to choose notation that helps keep concepts clearly distinguishable but at the same time is suggestive in that it makes apparent the connection between related objects, and overall notation that is convenient and easy to follow. As reference I suggest looking e.g. the papers of Andreas Maurer ("Algorithmic Stability and Meta-Learning" and "A Second-order Look at Stability and Generalization") or the statistical learning literature (papers/surveys by Massart, Boucheron, Lugosi and others) which I consider good examples with regards to the mathematical language that notation that hey use. In particular, I recommend the convention of using capitals (e.g. $X$ or $Y$ or $Z$) to denote random variables, using their lower-case counterparts ($x$, $y$, $z$ respectively) to denote arbitrary possible values of these random variables, and using calligraphic capitals ($\mathcal{X}$, $\mathcal{Y}$, $\mathcal{Z}$) for the sets of all possible values of these variables (i.e. their "spaces"). It makes perfectly clear the distinction between the different categories of objects (a random quantity, a fixed but arbitrary quantity, or the set of all possible values, respectively). I also recommend a separate kind of notation for the distributions of random variables, like $P$ (for "probability") or $\mu$ (must come from "measure" and is used often to denote a probability measure), and using subscripts for different $P$'s or $\mu$'s (e.g. $P_t$ or $\mu_t$ for the distribution of task $t$). Then if needed you could choose suitable notation for the set of all probability distributions over a given space, and so on.

Last note on notation choice: It could be good to make the distinction between expected loss and empirical loss visually apparent, say $L(h,\mu_t)$ and $\hat{L}(h,S_t)$ where the "hat" in the latter indicates the sample average.

L.124-125: "More recent frameworks include PAC-Bayes theory [...]" (since the acronym has been declared before)

L.127: " [49, 31, 19, 44]" (i.e. reference [44] belongs in this context as well, as do the works of Parrado-Hernandez et al. (2012) "PAC-Bayes bounds with data dependent priors" and Ambrolarze et al. (2006) "Tighter PAC-Bayes bounds" arguably.)

L.128: " [31, 33, 41]" (i.e. reference [41] is relevant here, which supersedes [45] with respect to tightening the bound values) and for the works that generalize the framework perhaps the list [13, 14] should be enlarged with Rivasplata et al. (2020) "PAC-Bayes Analysis Beyond the usual Bounds" at least.

Note that there is an important difference between a citation like [27] for "FMTL" and a citation like [23] for "FTL" since in the former case "FMTL" was introduced in the given citation [27] while in the latter case "FTL" existed in the literature before [23]. Hence, to make this difference apparent, I suggest writing "see e.g. [23]" if you want to use [23] as a reference for FTL. I am not objecting [23] as a suitable reference to look up the definition of FTL, in fact I believe it is, what I am suggesting is to use "see e.g." in order to signal that [23] is a good place to look up "FTL" while there are (many) others. Please make sure this distinction is followed for other references throughout the paper.

L.176: Replace "by some" with "by a copy" and clarify the distributional assumption of the $z'_i$ for the sake of clarity. Unless these data points are not random. But if $z_1, ..., z_m$ are random, as it sees from the problem formulation (L.86), then it makes sense that   $z'_1, ..., z'_m$ should also be random, and probably they are independent from $z_1, ..., z_m$ but coming from the same distribution.

L.180-182 (Definition 1): I don't see the point of "differ-by-one" (nor the corresponding acronym) since there is no other notion of change in the dataset in your paper (i.e. there is no "differ-by-two" or "differ-by-three"), On the other hand, as is well known, analysing stability to changing two or three data points reduces to analysing change in one data point at a time by the triangle inequality, hence the stability to changing one datapoint is the fundamental notion here. I prefer to read "Algorithm $A$ is uniformly stable with $\beta>0$ with respect to a loss $L$ if ..." and then after Eq.(2) write that you define $\beta_{US}$ to be the minimal $\beta>0$ in the right hand side. Since it is specific to algorithm $A$, probably it is even better to define $\beta_{US}(A)$ as the minimal such $\beta$. There is something missing in the quantification of variables in this definition, since $z'_i$ are currently not quantified. Also, if this condition is for random $z_1, ..., z_m$ and $z'_1, ..., z'_m$ (i.e. they are random variables) then probably the requirement should be that the inequality holds almost surely. But it looks that your definition wants to require that the inequality holds for any arbitrary $z_1, ..., z_m$ and $z'_1, ..., z'_m$, in which case the notation choice is hot helping to distinguish random variables from the arbitrary values that they may take. [A notation choice that I recommend is using capital letters (e.g. $X$) for a random variable, the corresponding lower-case ($x$) for an arbitrary possible value of this random variable, and capital calligraphic ($\mathcal{X}$) for the space of all possible values of this random variable. If used consistently, this notation makes it crystal clear what category of object occurs at each place.] In case (as I suspect) this definition is meant to require the inequality for all arbitrary choices of $z_1, ..., z_m$ and $z'_1, ..., z'_m$, it could be more convenient to write Eq.(2) with a supremum on the left-hand side, and clarifying in the next line that the supremum is over all $m$-tuples $(z_1, ..., z_m)$ and $(z'_1, ..., z'_m)$ that differ at one entry, and over the other things ($z$ and $i$ and $\theta$). Also I'd like to point out that the role of the task $t$ is not apparent in Eq. (2) currently, which needs clarification.

L.183-184: Since there is no citation to which this Theorem 1 is attributed, it could be assumed that you are implying that this theorem is yours? Could be better to write "The following result [22, Theorem X.Y] establishes a relationship between uniform stability and generalization in expectation." (Note that I am suggesting to be explicit about which theorem in [22] you are citing.)

L.188: This is a good place to discus the outline of the proof of Theorem 1, and insert a cross-reference to where the proof is given in full detail.

L.196: Is "sample $s$" a single datapoint? Ir is $s$ a list of datapoints? This needs to be clarified. Correspondingly, the distribution $\mathcal{D}$ should be over the space of a single data point or over some product space for tuples of data points, this needs to be said explicitly for the sake of clarity.

L.199: "the following holds simultaneously for all" (i.e. insert "simultaneously")

Note that the right hand side of Eq.(6) shows that you are using the form of McAllester's bound with the sharp dependence on the number of examples $l$, and this sharp dependence is due to Maurer (2004) "A note on the PAC-Bayesian theorem" (McAllester's bound was similar but with a mildly weaker dependence on the number of examples). On this note, the bound of McAllester that you are implying by Eq.(6) was due to McAllester (1999) "PAC-Bayesian Model Averaging" (COLT 1999), while the reference [34] in your list of references is McAllester (1998) "Some PAC-Bayesian Theorems" (it appeared first in COLT 1998, and then next year it also appeared in the journal Machine Learning). The point is that the correct year for [34] is 1998. But the bound that you are implying in Eq.(6) is the one given in "PAC-Bayesian Model Averaging" (with the sharp dependence on $l$ as per Maurer (2004)).

L.206: I miss the point of defining this marginal distribution $\mathcal{D}_S$ over datasets of size $m$. Should not such marginal distribution be $\mathcal{D}^m$ (i.e. the product of $m$ copies of the distribution $\mathcal{D}$)?

L.209: This line is puzzling. In PAC-Bayes bounds the quantity being bounded is an expectation of a loss of the form $L(h_{\theta},S)$ where the expectation is over the distribution of $\theta$ and over the distribution of $S$. In your Eq.(7) the two arguments of this loss are coupled, as both depend on $S$. Please elaborate on how to reason about this? There are some conditional expectations that have not been made explicit? Does the PAC-Bayes bound still hold in this case?

L.215: The assumption is that $0 \leq L(h,z) \leq 1$ for any $h$ in the hypothesis space for the given problem, and any $z$ in the example space. The current formulation is unnecessarily overloaded.

L.238-239: The notation "$s$" was used for a sample in the previous page, while here in this page it is used for the marginal variance parameters of a Gaussian with diagonal covariance. Could we use a different symbol? Also, it would be good to be explicit about the dimensionality of $\theta$ and then it will be clear the dimension of the space over which we have these Gaussian distributions.

L.241: Explain the symbol $\odot$ used in Eq.(10) which I assume is a Hadamard product?

L.251-252: Could the authors provide an appendix with details for the extension to mini-batches of tasks?

L.252-253: "stochastic gradient descent (SGD) and gradient descent (GD)"

Caption of Table 1: "We also report the meta-test loss (CEL_s) for all methods."
Also, may we see the definition of the acronym CEL (and a suitable reference)?

L.275: "cross-entropy" (with hyphen)

L.284: Was this notation declared? May be I missed it, but it would not harm to clarify (write explicit) that N is the neural network and  the norm with subscript F is the Frobenious norm.

L.335-336: Insert a reference to support the claim that "it may cause memorization" which refers to learning the prior, was this something reported by [58]? Or if this was something that was observed in your own experiments, then explain this fact with accompanying discussion.

L.345: "meta-learning: PAC-BUS" (i.e. replace the comma with a colon)

L.367: What are "broader PAC-Bayes bounds"? I did not follow the meaning of "broader" in this sentence (while the meaning of "tighter" is clear, of course).

L.368-369: "using a PAC-Bayes bound for unbounded loss functions for the meta-generalization step"


L.296: "through the continuous-bag-of-words GloVe embedding [38] into dimension d = 50 to generate samples"

Would like for all my feedback above to be addressed in the author response.

**Time Spent Reviewing:**

7 hours

---

> ### Author Response · Authors · 2021-08-10
> **Response to Reviewer mTwB**
>
> Thank you very much for your valuable and thorough feedback. We appreciate your positive comments on our work and suggestions for improvement. We address each of your points below.
>
> ### Significance:
> We provide a very short outline of the proof method for Theorem 3 at the beginning of Section 4.2 with important steps mentioned at the ends of Sections 4.1.1 and 4.1.2. However, we agree that this can be expanded upon. We will provide an outline in the revision and a detailed proof in the Appendix. We provide an outline here which is consistent with your brief outline: \
> First, we make use of the PAC-Bayes framework to bound the expected training loss achieved after running an algorithm A in terms of its empirical counterpart. We begin with the statement of Theorem 2, a general statement of the McAllester PAC-Bayes bound. We let $\mathcal{D}$ denote the marginal distribution over datasets of size $m$ and note that sampling from $\mathcal{D}$ is equivalent to sampling a task $t \sim \mathcal{T}$ and then sampling $S \sim \mathcal{Z}^m_t$. If we are provided with a set of training datasets (drawn i.i.d. from $\mathcal{D}$), we can (with high probability) bound the expected training loss on any training dataset drawn from $\mathcal{D}$.  The resulting PAC-Bayes bound is presented in Inequality (7). \
> Next, we establish a bound on the true expected loss for a new task after running algorithm A on a training dataset corresponding to the new task in terms of the expected training loss. Beginning with the statement [22, Theorem 2.2], let algorithm A be deterministic; this results in Inequality (3). We then replace the expected loss on a training dataset with the expectation with respect to $\theta \sim \Theta$ of this loss. In addition, we take the expectation over $t \sim \mathcal{T}$. The result is presented in Inequality (4) and holds under the same assumptions as Theorem 1. \
> Lastly, we combine the resulting bounds to achieve a bound on the expected true loss (after running algorithm A) in terms of the empirical training loss. Let algorithm A be $\beta_{US}$ uniformly stable and note that the RHS of Inequality (7) now satisfies the assumptions of Theorem 1 and can be plugged into the RHS of (4). Since the PAC-Bayes bound held with high probability, the resulting bound (Inequality (7)) also holds with high probability under the assumptions of both Theorem 1 and Theorem 2.
>
> ### Editorial feedback:
> We are grateful for your thorough editorial feedback. We address each of your comments below.
>
>
> Title: We think this is a good idea and would be happy to make this change.
>
>
> L.8-9, L.19, L.23, L.36, L.43, L52, L.60-61, L.68-69, L.124-125, L.199, L.215, L.252-253, L.275, L.296, L.345, L.368-369: We agree with each of these editorial suggestions and will make these changes in the revision.
>
>
> L.48, work of Shalev-Shwartz et al. and Poggio et al., L.127, L.128: Thank you for the additional references. We will add these citations and mention them in the related work.
>
>
> L.50-51: You are correct here that the definition uniform stability is not consistent between [11] and [22]. We will rephrase as suggested to "[22] demonstrates that limiting the number of training epochs of a gradient-based learning algorithm leads to stability in expectation."
>
>
> Section 2, notation choice: Thank you for the example papers and suggested notation changes. We will modify the notation to remain consistent between samples, random variables, spaces of possible values, and distributions over such spaces. We will also change empirical loss to $\hat{L}$ in the revised version of the paper; this will help with readability.
>
>
> Citation [27] vs [23]: Thank you for this suggestion. We will ensure the distinction between introduction and reference is made clearly throughout the paper in the revision.
>
>
> L.176: Here we do not mean that $z_i$ and $z_i'$ are random but some arbitrary element of the set of all possible $z$. We will clarify this in addition to the suggested notation modifications.
>
>
> L.180-182: We mentioned "differ-by-one" uniform stability to distinguish specifically from the common definition of "leave-one-out" uniform stability. However, as you mention, we only make use of this single definition, so it is not necessary to add the extra notation. Thank you for the suggestion --- we will modify this in the revision. We also appreciate your comments on the notation here. You are correct in assuming that we mean the inequality must hold for arbitrary choices of z, S, i, and $\theta$. A supremum is a good choice to make this more clear in addition to the notation changes you suggest here and above (i.e. $\mathcal{X}$ for the space of all possible values of the random variable $X$ and $x$ for an arbitrary possible value of the random value). We will make these clarifications in the revision. We will also remove the conditioning on a task $t$ here, as it is not necessary in this definition.
>
>
> L.183-184, L.188: Theorem 1 is a Corollary of [22, Theorem 2.2] where we let algorithm A be deterministic. We will clarify this in the revision and will include a reference to the existing discussion (Appendix L.48-L.53) of using deterministic algorithms instead of randomized. In addition, we will provide a proof outline of Theorem 1.
>
>
> L.196, L.206: We believe the confusion here is partially due to our notation (which we will revise). In this case, sample $s$ is a dataset and $\mathcal{D}$ is a distribution over a space of tuples of data points. We define this marginal distribution over datasets in order to employ the PAC-Bayes bound on the expected training loss (over $\mathcal{D}$) in terms of the empirical training loss. This is needed in the first step of the proof of Theorem 3, which we outline above. \
> We mention the purpose for this marginal distribution briefly in the outline of Theorem 3 above and will clarify this section and notation in the revision.
>
>
> Citation for Eq. (6): Thank you for pointing this out. We will include the Maurer (2004) reference here and ensure we are correctly citing the 1998 and 1999 McAllester papers.
>
>
> L.209: A similar PAC-Bayes bound after adaption is provided and explained in [58]; see specifically Equation (18) and the proof of [58, Theorem 1]. The statement holds with an application of the data-processing inequality (ref [5] of the appendix). Specifically, let $\tilde{\theta} = A(\theta, S)$, and let $\tilde\Theta(S) := A(\Theta, S)$ be the distribution over hyperparameters after algorithm $A$ adapts $\theta \sim \Theta$ with $S$. Next, consider the KL divergence term between the adapted prior with the adapted posterior: $E_{S\sim\mathcal{D}} KL(\tilde\Theta(S) || \tilde\Theta_0(S)) \leq E_{S\sim\mathcal{D}} KL(\Theta || \Theta_0) = KL(\Theta || \Theta_0)$. We will make this dependence clear in the full proof of Theorem 3 and include references for this step in the revision.
>
>
> L.238-239: We agree that the notation $s$ here is conflicting and we will modify it in the revision version of the paper. We will also add the dimension of theta here.
>
>
> L.241: This is indeed the Hadamard product. We will introduce this notation in the revision.
>
>
> L.251-252: We will include an appendix with the extension to mini-batches of tasks in the revised version of the paper.
>
>
> Caption of Table 1: We agree with your editorial suggestion here and will revise this. We define CEL$_s$ (the softmax activated cross-entropy loss) in L.275 but will also include it in the caption for clarity.
>
>
> L.284: Declaration of this notation is indeed missing from the body of the paper. We will add this in the revised version of the paper and point to Appendix A.5 where we elaborate on the constraint.
>
>
> L.335-336: In this sentence, we mean that the prior may memorize the data used to train it. However, this does not necessarily mean that the methods would then be more likely to memorize in training the posterior. We do not have observations which would support such a fact. We will clarify this in the revision.
>
>
> L.367: By “broader PAC-Bayes bounds”, we are referring to PAC-Bayes bounds which have a different set of assumptions to the McAllester PAC-Bayes bound [34] and may allow for non-vacuous bounds in broader settings, e.g., PAC-Bayes bounds for unbounded losses [21] (mentioned on L.368-369). We will clarify this in the revision and reword to "PAC-Bayes bounds with less restrictive assumptions (e.g., bounds for settings with unbounded losses)".

---

### Official Review · Reviewer_wSzt · 2021-07-15

**Rating:** 6
**Confidence:** 4

**Summary:**

The paper provides generalization bounds for meta-learning approaches. The result combines two frameworks: uniform stability of gradient descent based learning algorithms that cover guarantees at the base classifiers level and PAC-Bayes theory for obtaining results at the meta level. The objective is to have bounds that become tighter when the base learner adapts quickly. The framework proposed allows the authors to develop new algorithms and regularization scheme. An experimental evaluation is conducted on a toy problem and standard benchmarks.

**Limitations And Societal Impact:**

The authors discuss some limitations of the approaches in the conclusion, in particular the convergence properties of the algorithms provided.
They also provide a section on the broader impact of the work.

The work proposed in this paper is mainly fundamental and does not imply some direct negative societal impact. Actually, by trying to provide some generalization guarantees, the papers contributes to a better general understanding of the field.


**Main Review:**

Originality
---------
Understanding fewshot learning methods is still challenging and results going in this direction are important. The proposed contribution combines an existing uniform stability framework with a classic version of the PAC-Bayes framework. This is not particular new, the novelty lies here in the application to the fewshot learning framework.
In my opinion, the novelty is moderate.



Quality
------
The global scope is of ok. The authors have completed their claims by complementary results in Appendix.
I have however the following reservations:
-The uniform stability framework of (Hardt et al., ICML'16) is initially presented for randomized algorithms, so the validity of the contributions for non randomized models has to be justified. See the remarks in the comments below.
-The PAC-Bayes term of Eq.(7) includes only the number tasks and does not include the size of the samples.
-Comparisons to over PB-based frameworks would have strengthened the experimental part.


Clarity
------
The paper is globally clear.
Few additional lines for describing a bit clearly the fewshot framework (meta-learning, meta-testing, k-way N-shot) would certainly help an unfamiliar reader to better understand some concepts presented in the paper.



Significance
----------------
The basis of the work relies on existing techniques. The result obtained in Thm 3 is relatively direct from the considered frameworks. The contribution represents another step in considering PAC-Bayes for few shot learning, but the result in my opinion is not particularly significant.
In particular, the result does not exhibits how the combination between tasks and the number of instances can lead to learn with few examples: even with the increasing number of training data, the number of tasks still need to be important. From the bound, it seems that the number of tasks considered should be very important.

From the practical evaluation, the bounds seem informative, but comparisons to other settings would also be helpful. I guess here that the quality of $\Theta_0$ is essential and I did not see much details on how to find this initialization, which would also be interesting for reproducing the results.

Thm1 and deduced results has to be refined with respect to the correct framework.


Comments
------------

-About Thm1: If I am not mistaken, the result in Thm1 coming from (Hardt et al., ICML'16 [22]) is valid for randomized algorithms and then an expectation over A must be added in the definition. Same for Equation 4. So the application to non stochastic algorithms must be better justified.

In the paper, the authors consider theequivalent  following definition of stability
$$
sup_{S,S^i} sup_z sup_\theta [L(h_A(\theta,S), z) - L(h_A(\theta,S^i), z)] \le \beta_{US},
$$

while, the definition of (Hardt et al., ICML'16 [22]) corresponds to
$$
sup_{S,S^i} sup_z E_\theta [L(h_A(\theta,S), z) - L(h_A(\theta,S^i), z)] \le \beta_{US}
$$

The definition considered in the paper implies the definition of (Hardt et al., ICML'16 [22]). I hope this is what the authors expect, but at least a comment on the change of definition should be made.

However, in Thm 2.2 of (Hardt et al., ICML'16 [22]), it is shown that
$$
 E_{S,\theta} L(h_{A(\theta,S)}, T) \le E_{S,\theta} L(h_{A(\theta,S)}, S) + \beta_{US}.
$$

So actually, an expectation is missing.

This expectation is missing when one wants also to prove Thm3. That being said, a more formal justification of Thm3 (i.e. with a proof) could be useful maybe.

The validity of the approach for gradient descent (and not SGD) should also be discussed accordingly.


-Thm2 uses the classic McAllester version. It could be interesting to obtain other frameworks to obtain tighter results that may be more informative for the few-shot learning context

-Equation 7: here, I guess we assume that each sample S_i has the same number of examples. As far as I can understand, the result, the number of examples is not involved in the PAC-Bayes term, which makes this result similar to frameworks using an hyperprior model.


-In Table 1: the only PAC-Bayes bound comes from reference [3], but it would be interesting to compare to other PAC-Bayes results.

-l.223-226, distribution D_{S_{ev}} is introduced here but the justification of this distribution is a bit fuzzy. Few additional lines to explain it would be welcomed here.


-About the result. Other recent meta-learning frameworks have tried to propose bounds in O(l\times (m+n)) [Du et al., Few-Shot Learning via Learning the Representation, Provably, 2020; Tripuraneni et al., Provable Meta-Learning of Linear Representations, 2020].  This could be a strong result to obtain such bounds in a Pac-Bayes context.

I also would have appreciated a discussion on the expressiveness of the framework with respect to other PB-settings for fewshot/meta-learning learning, beyond the computational issues: what is more general, less/more expressive, can the proposed framework be derived from existing frameworks by choosing particular hyperprior for example?

-Lines 272-273 "Notably, this problem is challenging for losses which are convex in the parameters of the network (since the decision boundary between classes is nonlinear)" - I am not sure to understand the formulation in general, since what matters is non linearity and not especially non-convexity. It maybe easier to have non-linear formulations with non-convex models, but the formulation should be rewritten here.

-In the related work, the authors should consider and discuss the following reference that is one the first mixing the concepts of stability and PAC-Bayes:
*Ben London: A PAC-Bayesian Analysis of Randomized Learning with Application to Stochastic Gradient Descent. NeurIPS 2017.
This is not exactly the same setting, but this paper is important regarding the topic

-In appendix, lines 18-23, about the fact to use the training data is necessary to have guarantees on performance. I may have missed something, but this points seems to require more justification, in particular why not having another validation set is not sufficient? The learned models become independent from S. The bound might be over-optimistic.
Then for the validity of the PB bound this raises questions because the optimization of \theta should be justified to not depend on S.


-----------
After rebuttal
-----------
The authors did a very important work in order to bring more justification and additional information.
From the feedbacks provided, I am now convinced that the work is correct and of quality.
Adding a study on the impact of the parameters of algorithm on the bound and the behavior of the model is for me very interesting in the context of the work.
The paper needs now to be updated according to the comments provided to improve the clarity of the paper.
The reservations on the work, I have are that the potential increasing of the bound when training data increases and the possibility to use large validation sets in a few-shot learning contexts, anyway in a multi-task setting this point would be less important.


**Time Spent Reviewing:**

12

---

> ### Author Response · Authors · 2021-08-10
> **Response to Reviewer wSzt**
>
> Thank you very much for your valuable and thorough feedback. We address each of your points below.
>
> ### Originality:
> We believe that the combination of the two generalization frameworks (PAC-Bayes and algorithmic stability) is a key contribution and a strength of our work. This combination is particularly well-suited to the goal of meta-learning: training a base-learner to learn as quickly as possible on new data. This regime (where the base learner only takes a few gradient steps) is the regime in which uniform stability bounds are tightest; thus, we employ stability bounds at the base-learning level. We then see the problem of generalizing to new tasks in the meta-learning setting as analogous to generalizing to new data in the supervised learning setting. Due to the recent success of PAC-Bayes bounds in providing strong generalization guarantees for neural networks in the supervised learning setting, we employ it at the meta-level. As our numerical results demonstrate, employing PAC-Bayes bounds at both levels ([3, 58]) results in looser bounds. In addition, employing algorithmic stability bounds at both levels (such as in [32]) results in restrictive assumptions since the meta-learning algorithm must be uniformly stable. Thus, we believe this novel combination of generalization framework solves a significant challenge which could not be solved by an individual bound. We also point to the other reviewers, who all note that the combination of these two frameworks for a meta-learning bound is novel.
>
> ### Quality:
> Use of non-randomized algorithms: We present Theorem 3 with non-randomized algorithms to reduce the computation time for the upper bound. If an expectation over algorithm A is included in the upper bound, we would require another sample complexity bound in addition to the one we have for the expectation over $\theta$ (L.257-258). We discuss this briefly on L.48-53 of the Appendix and will include a discussion of this when we introduce Theorem 1 as well as a reference to the Appendix for further justification. \
> The other points are addressed below in the response to comments section.
>
> ### Clarity:
> We will add further explanation of the the few-shot learning setting and expand upon the formulations for the examples to help readability and clarity. Thank you for the suggestion.
>
> ### Significance:
> A small number of tasks can result in a larger PAC-Bayes regularization term and loosen the PAC-BUS bound. This is analogous to a key challenge with generalization bounds in the standard supervised learning setting -- achieving tight bounds with a small number of samples. An advantage of the PAC-Bayes framework is that it allows for specification of a prior, which can improve the bound when there are a smaller number of tasks. We briefly describe the training method for the prior used in Examples 5.1 and 5.2 in Appendix A.8.1 and A.8.2 respectively. In both cases, we use use held-out data to train a prior. We will expand upon the details in the appendix and the in the main body of the paper in the revision in addition to addressing the point above.
>
> ### Comments:
> Thm 1: Theorem 1 is a Corollary of [22, Theorem 2.2] where we let algorithm A be deterministic. The proof is nearly identical to the one provided in [22] and we will provide a sketch in addition to clarifying this point before presenting Theorem 1. The definition of stability in [22] is indeed different from the one presented in [11]. As suggested, we will clarify the distinction before presenting Theorem 1 in the revision. \
> Additionally, we provide an outline of the proof of Theorem 3 in the significance section of the response to reviewer mTwB. We will include an outline of the proof in the body of the paper and a complete proof in the appendix.
>
>
> Thm 2: We are making use of the Maurer version of the PAC-Bayes bound -- we will clarify this in the revision. In addition, switching out the PAC-Bayes bound for more recent ones would be straightforward with our framework, i.e. the quadratic PAC-Bayes bound [45] and the PAC-Bayes-$\lambda$ bound [53]. We mention this briefly in L.193-194 but will expand upon this in the revision.
>
>
> Equation 7: In practice we keep the number of samples in each $S_i$ consistent. However, this is not required for the guarantee to hold and nor do we assume it in the derivation.
>
>
> Expressiveness of framework: We don’t believe the proposed framework can be derived from existing frameworks by choosing a particular hyperprior. Such a result would be interesting in unifying frameworks and an interesting direction for future work. One could ask similar questions for the relationship between PAC-Bayes and algorithmic stability in the standard supervised learning setting; we are not aware of a fully general answer to this even in the standard supervised learning setting.
>
>
> Table 1: We also compare to the bound presented in [58]. This paper presents a PAC-Bayes meta-learning bound which improves on many of the drawbacks of other PAC-Bayes bounds for meta-learning, such as the computation time required to evaluate the upper bound. We note this in the related work on L.153-156. We believe that this, along with [3], provides an effective comparison between existing bounds for PAC-Bayes meta-learning bound and our bound.
>
>
> L.223-226: We define the marginal distribution $\mathcal{D}$$\_S$  over datasets to provide a PAC-Bayes bound on the expected training loss in terms of the empirical training loss. This is required in the proof of Theorem 3 (an outline of the proof is provided in the significance section of the response to reviewer mTwB). We follow a similar procedure when deriving the bound which allows for validation data, Inequality (9). We define $\mathcal{D}$$\_{S_{ev}}$ to provide a PAC-Bayes bound on the expected evaluation loss in terms of the empirical evaluation loss. We then use this bound to relate the expected training loss to the empirical evaluation loss (see Appendix A.1.2 for more details on this step). We can then follow the last two steps of the proof of Theorem 3 to produce Inequality (9). We will expand upon the justification for this in the derivation of Inequality (9).
>
>
> Other recent meta-learning frameworks:
> Thank you for mentioning these references. This is a good idea, and we will explore such risk bounds. However, we do not believe these are necessary for the bounds to be effective in practice. All upper bounds are valid even if posterior over initializations is not locally or globally optimal. \
> We agree that arguments for the generality or expressiveness of the bounds would be convincing. However, we believe it would be difficult to theoretically argue that certain bounds are more general than others. Indeed, even in the supervised learning setting, it is not always the case that PAC-Bayes bounds are more general or tighter than uniform stability bounds, or vice versa. We believe that a qualitative argument in combination with our empirical results – which do show that the bound can be tighter in practice – is convincing in lieu of a quantitative argument.
>
>
> L.272-273: We agree that non-linearity is a more informative property here than non-convexity. We will modify this problem formulation in the revision to improve clarity.
>
>
> Additional related works: Thank you for pointing out these references, we will include them in the related work.
>
>
> Appendix L.18-23: To combine the bounding frameworks, we require a bound on $E_{\mathcal{T}, \mathcal{Z}^m, \Theta}L(h_{A(\theta, S)}, S)$. In the few-shot learning case, we bound this using an empirical loss which includes validation data. Equation (A.4) establishes a relationship between $E_{\mathcal{T}, \mathcal{Z}^{m+n}, \Theta}L(h_{A(\theta, S)}, S_{ev})$ and $E_{\mathcal{T}, \mathcal{Z}^m, \Theta}L(h_{A(\theta, S)}, S)$ which we use form such a bound: Inequality (A.6). \
> The only relationship we have between $E_{\mathcal{T}, \mathcal{Z}^n, \Theta}L(h_{A(\theta, S)}, S_{va})$ and $E_{\mathcal{T}, \mathcal{Z}^m, \Theta}L(h_{A(\theta, S)}, S)$ is given by Inequality (4) when algorithm A is $\beta_{US}$-uniformly stable (since
>  $E_{\mathcal{T}, \mathcal{Z}^{n}, \Theta}L(h_{A(\theta, S)}, S_{va}) = E_{\mathcal{T}, \Theta}L(h_{A(\theta, S)}, \mathcal{Z})$). It’s not clear how to combine combine the generalization frameworks in this case. We will elaborate on the discussion about this in the revision.

---

> > ### Comment · Reviewer_wSzt · 2021-08-25
> > **Thanks for the answers and additional questions**
> >
> > Dear authors,
> >
> > Many thanks for the feedback to each of my comments. I appreciate the effort.
> >
> > I have still the following remarks/issues:
> >
> > -about the proof of Thm3. Thanks for the precisions in the answer to reviewer mTwB. However, to be honest, as a reviewer I cannot be confident without seeing the whole proof, in particular in the case of a main result. Maybe using multiple answers (ie multiple comments to the post), you may be able to fully write the proof.
> >
> > -Since one of the goal of the contribution is to consider gradient descent-based algorithms, I would have appreciated to see the evolution of the bounds and the performances of the algorithms with different iterations and learning rates. This would be interesting to see if for bad parameters the bounds tend to be high indicating the presence of a bad situation. This would be also more in coherent with the context of stability considered here.
> >
> > -In the bounds of Table 2, the bound is increasing when the number of examples per class gets larger. This is a bit surprising and deservers maybe a deeper discussion. I was not able to completely understand the arguments given to reviewer Q1zC.
> > I am not able to understand when the 3rd term should increase, for fixed parameters and fixed number of validation data, the term should decrease when $n$ increases, no?
> > I may guess that in practice the bound becomes higher because of different gradient-descent parameters (nb of iterations, learning rate), is it right?
> > But, then this does not necessarily help  to explain few-shot learning: with a bound in $O(\frac{1}{\sqrt{l}} + \frac{m}{m+n})$, with small $n$ regarding $m$, we do not explain easily generalization.
> > I am actually wondering if this result is not more adapted to the classic multi-task setting where you can have large samples per tasks.
> >
> > About the validation set: actually in a few-shot learning setting, I would have expected to consider additional tasks in the validation set (as done in many few-shot learning papers), and not additional samples. This contradicts a bit with the notion of few-shot where the number of examples is small, but I may have wrongly understood your setting.
> >
> > -About the comparisons to other methods, I think a comparison to the PAC-Bayes approach of paper [46] would make sense. Maybe their bound will be looser as you argued in your paper, but maybe in practice their additional divergence measures could help the algorithm to better focus on particular distributions and adapt better in some situations. And if their bounds are larger than yours, this is also a good point for your paper.
> >
> > -A last editing comment, when you present the notion of stability, it is maybe more appropriate to directly consider a formulation adapted to your setting, and provide additional proofs (of Thm1 for example), which would avoid to mix different things and involve some confusion.
> >
> > Thanks for reading this long message.

---

> > > ### Author Response · Authors · 2021-08-30
> > > **Response to additional comments**
> > >
> > > Thank you very much for your additional feedback. We appreciate the opportunity to clarify any lingering questions. We address each of your points below (in two messages). Please excuse the minor notation changes (expected value $\mathbb{E} \rightarrow E$, marginal distribution over datasets of size $m$ $\mathcal{D}_S \rightarrow D_S$, distribution over tasks $\mathcal{T} \rightarrow T$, distribution over samples $\mathcal{Z} \rightarrow Z$) that were made so that equations compiled correctly in this reply.
> > >
> > > We will begin with the proof of Theorem 3 and also prove a Corollary to [Theorem 2.2, 22] which we use in the proof, for completeness. Note that the proof steps for Theorem 3 presented here are also presented in the paper. Specifically, step 1 is described in Section 4.1.2, step 2 is presented in 4.1.1, and step 3 is in Section 4.2. We have made some clarifications in this proof and will revise the paper to include this complete proof in the appendix as well as clarify the proof path in the body of the paper.
> > >
> > > ### Proof of Theorem 3:
> > > **Step 1:**
> > > Define marginal distribution $D_S$ over datasets of size $m$ and note that sampling $S \sim D_S$ is equivalent to first sampling $t\sim T$ and then sampling $S \sim Z^m_t$. Consider the following
> > > \begin{equation}
> > > E_{t \sim T} \ E_{S \sim Z^m_t} \ E_{\theta \sim \Theta} \ L(h_{A(\theta, S)}, S) = E_{S\sim D_S} \ E_{\theta \sim \Theta} \ L(h_{A(\theta, S)}, S)
> > > \end{equation}
> > > where $A(\theta, S)$ is any deterministic algorithm. We make the following definition for simplicity: $f_A(\theta, S):= L(h_{A(\theta, S)}, S)$; thus we equivalently consider the following:
> > > \begin{equation}
> > >     E_{S\sim D_S} \ E_{\theta \sim \Theta}\ f_A(\theta, S).
> > > \end{equation}
> > > Assume that loss function $f_A$, which maps a set of parameters $\theta$ and a dataset $S$ to a real number, satisfies Assumption 1 (i.e. is bounded in $[0,1]$). Due to [34] (with the tightening step from [R1] for $l \geq 8$), we have the following PAC-Bayes guarantee on the expected training loss achieved after running an algorithm A:
> > >
> > > For data-independent prior distribution $\Theta_0$, loss function $f_A$ which satisfies Assumption 1, $l \geq 8$, and $\delta \in (0,1)$, with probability at least $1-\delta$ over a sampling of $\textbf{S} = \\{S_1, S_2, \dots, S_l\\} \sim D_S^l$, the following holds for all $\Theta$:
> > > \begin{equation}
> > >     E_{S\sim D_S} \ E_{\theta \sim \Theta}\ f_A(\theta, S) \leq \frac{1}{l} \sum_{i=1}^l \ E_{\theta \sim \Theta} \ f_A(\theta, S_i) + \sqrt{\frac{D_{KL}(\Theta \| \Theta_0) + \ln\frac{2\sqrt{l}}{\delta}}{2l}}. \ \ \ \ \ \ \ (1)
> > > \end{equation}
> > >
> > > **Step 2:**
> > > Let algorithm $A$ be $\beta_{US}$-uniformly stable. For a fixed task $t \sim T$ we have the following (proof for corollary to [Theorem 2.2, 22] is below):
> > > \begin{equation}
> > >     E_{S \sim Z_t^m} \ E_{\theta \sim \Theta} \ L(h_{A(\theta, S)}, Z_t) \leq E_{S \sim Z_t^m} \ E_{\theta \sim \Theta} \ L(h_{A(\theta, S)}, S) + \beta_{US}.
> > > \end{equation}
> > > Take the expectation over $t \sim T$. We then have:
> > > \begin{equation}
> > >     E_{t \sim T} \ E_{S \sim Z_t^m} \ E_{\theta \sim \Theta} \ L(h_{A(\theta, S)}, Z_t) \leq E_{t \sim T} \ E_{S \sim Z_t^m} \ E_{\theta \sim \Theta} \ L(h_{A(\theta, S)}, S) + \beta_{US} \ \ \ \ \ \ \ (2)
> > > \end{equation}
> > > since $E_{t \sim T} \ \beta_{US} = \beta_{US}$. This establishes a bound on the true expected loss for a new task after running algorithm A on a training dataset corresponding to the new task.
> > >
> > > **Step 3.**
> > > We rewrite (1) using the definition for $f_A$ and the noted sampling equivalence for $S \sim D_S$:
> > > \begin{equation}
> > >     E_{t \sim T} \ E_{S \sim Z^m_t} \ E_{\theta \sim \Theta} \ L(h_{A(\theta, S)}, S) \leq \frac{1}{l} \sum_{i=1}^l \ E_{\theta \sim \Theta} \ L(h_{A(\theta, S)}, S_i) + \sqrt{\frac{D_{KL}(\Theta \| \Theta_0) + \ln\frac{2\sqrt{l}}{\delta}}{2l}}. \ \ \ \ \ \ \ (3)
> > > \end{equation}
> > > Note that this provides an upper bound on the first term of the upper bound of (2) when algorithm $A$ is $\beta_{US}$ stable. Thus we have the following by plugging (3) in the RHS of (2):
> > >
> > > For data-independent prior distribution $\Theta_0$, loss $L$ which satisfies Assumption 1, $l \geq 8$, algorithm $A$ which is $\beta_{US}$-uniformly stable, and $\delta \in (0,1)$, with probability at least $1-\delta$ over a sampling of $\textbf{S} = \\{S_1, S_2, \dots, S_l\\} \sim D_S^l$, the following holds for all $\Theta$:
> > > \begin{equation}
> > >        E_{t \sim T} \ E_{S \sim Z_t^m} \ E_{\theta \sim \Theta} \ L(h_{A(\theta, S)}, Z_t) \leq \frac{1}{l} \sum_{i=1}^l \ E_{\theta \sim \Theta} \ L(h_{A(\theta, S)}, S_i) + \sqrt{\frac{D_{KL}(\Theta \| \Theta_0) + \ln\frac{2\sqrt{l}}{\delta}}{2l}} + \beta_{US}.
> > > \end{equation}
> > > QED.
> > >
> > >
> > > ### Proof of Corollary to [Theorem 2.2, 22]
> > > For completeness, we will also provide a proof for a corollary to Theorem 2.2 from [22] (a uniform stability bound for deterministic algorithms using the expected loss over $\theta \sim \Theta$) which we make use of in the proof for Theorem 3. Note that our analysis does not rely on algorithm $A$ being deterministic and the PAC-BUS bound can be derived for the case when $A$ is a randomized algorithm. However, we let $A$ be deterministic to improve the computation time required to estimate the upper bound after training.
> > >
> > > Consider the following notion of uniform stability analogous to [Def 2.1, 22], except using deterministic algorithms and distributions $\Theta$ over initializations: A deterministic algorithm $A$ is $\beta_{US}$-uniformly stable if for all samples $z$ in the set of possible samples, for all datasets $S, S'$ of samples of size $m$ such that $S$ and $S'$ differ by at most one sample, and a distribution over initializations $\Theta$, we have
> > > \begin{equation}
> > >     E_{\theta \sim \Theta} |L(h_{A(\theta, S)}, z) - L(h_{A(\theta, S)}, z)| \leq \beta_{US}.
> > > \end{equation}
> > >
> > > Corollary of [Theorem 2.2, 22]: Let deterministic algorithm $A$ be $\beta_{US}$-uniformly stable and fixed task $t \sim T$. Then
> > > \begin{equation}
> > >     E_{S \sim Z_t^m} \ E_{\theta \sim \Theta} \ L(h_{A\theta, S)}, Z_t) \leq E_{S \sim Z_t^m} \ E_{\theta \sim \Theta} \ L(h_{A(\theta, S)}, S) + \beta_{US}.
> > > \end{equation}
> > > _proof:_ (We follow analogous steps to the proof presented in [22]) Let $S = \\{z_1, z_2 \dots, z_m \\} \sim Z_t^m$ and $S' = \\{z_1', z_2' \dots, z_m' \\} \sim Z_t^m$ be two independent random samples and let $S^i = \\{z_1, \dots, z_{i-1}, z_i', z_{i+1}, \dots, z_m \\}$ be identical to $S$ except with the $i$th sample replaced with $z_i'$. Fix a distribution $\Theta$ over initializations. Consider the following
> > > \begin{align}
> > >     E_{S \sim Z_t^m} E_{\theta \sim \Theta} L(h_{A(\theta, S)}, S) & = E_{S \sim Z_t^m} E_{\theta \sim \Theta} \bigg[\frac{1}{m} \sum_{i=1}^m L(h_{A(\theta, S)}, z_i)\bigg] \\\\
> > >     & = E_{S \sim Z_t^m} \ E_{S' \sim Z_t^m} \ E_{\theta \sim \Theta} \bigg[\frac{1}{m} \sum_{i=1}^m L(h_{A(\theta, S^i)}, z_i)\bigg] \\\\
> > >     & = E_{S \sim Z_t^m} \ E_{S' \sim Z_t^m} \ E_{\theta \sim \Theta} \bigg[\frac{1}{m} \sum_{i=1}^m L(h_{A(\theta, S)}, z_i')\bigg] + \delta \\\\
> > >     & = E_{S \sim Z_t^m} \ E_{\theta \sim \Theta} \ L(h_{A(\theta, S)}, Z_t) + \delta
> > > \end{align}
> > > where
> > > \begin{equation}
> > >     \delta = E_{S \sim Z_t^m} \ E_{S' \sim Z_t^m} \ E_{\theta \sim \Theta} \bigg[\frac{1}{m} \sum_{i=1}^m L(h_{A(\theta, S^i)}, z_i') - \sum_{i=1}^m L(h_{A(\theta, S)}, z_i')  \bigg].
> > > \end{equation}
> > > We bound $\delta$ with the supremum over datasets $S$ and $S'$ differing by a single sample
> > > \begin{equation}
> > >     \delta \leq \sup_{S, S', z} E_{\theta \sim \Theta} [L(h_{A(\theta, S)}, z) - L(h_{A(\theta, S)}, z)] \leq \beta_{US}
> > > \end{equation}
> > > by the definition of uniform stability for algorithm $A$ above. QED.
> > >
> > > [R1] Andreas Maurer. A Note on the PAC Bayesian Theorem. _arXiv preprint arXiv:1307.2118_, 2004.

---

> > > > ### Author Response · Authors · 2021-08-30
> > > > **Continued response to additional comments**
> > > >
> > > > **Bounds and performances of the algorithms with different iterations and learning rates:** We agree that this can provide insight on the bound and have computed the following results as you suggest. We will include them in the revision. Note that in the interest of time, we have not used the sample convergence bound (ref. Sec. 4.3 and App. A.4) and present the results for a single sample $\theta \sim \Theta$. The true values of the upper bounds for MR-MAML [53] and PAC-BUS are unlikely to change by more than 5\% as the sample complexity bound does not loosen the guarantee very much. Additionally, as in the paper, we will not compute the sample convergence bound for MLAP-M [3] since it requires many more iterations to estimate). In the revision, we will compute the sample convergence bound as well as the mean after 5 trials as we do with other results.
> > > >
> > > > Below we present test losses for all methods for base-learning rates of $0.001$ to $10$ using $\\{1,3,10\\}$ adaptation steps.
> > > >
> > > > | MAML Test Loss, lr$_b$ = | 0.001 | 0.002 | 0.005 | 0.01 | 0.02 | 0.05 | 0.1 | 0.2 | 0.5 | 1 | 2 | 5 | 10 |
> > > > | --- | --- | --- | --- | --- | --- | --- | --- | --- | --- | --- | --- | --- | --- |
> > > > | Adaptation steps = 1 | 0.170 | 0.186 | 0.185 | 0.180 | 0.187 | 0.177 | 0.180 | 0.157 | 0.137 | 0.122 | 0.117 | 0.110 | 0.134 |
> > > > | Adaptation steps = 3 | 0.191 | 0.182 | 0.186 | 0.180 | 0.177 | 0.164 | 0.144 | 0.134 | 0.121 | 0.113 | 0.111 | 0.111 | 0.126 |
> > > > | Adaptation steps = 10 | 0.186 | 0.182 | 0.175 | 0.183 | 0.157 | 0.144 | 0.128 | 0.126 | 0.119 | 0.111 | 0.108 | 0.106 | 0.129 |
> > > >
> > > > | MLAP-M Test Loss, lr$_b$ = | 0.001 | 0.002 | 0.005 | 0.01 | 0.02 | 0.05 | 0.1 | 0.2 | 0.5 | 1 | 2 | 5 | 10 |
> > > > | --- | --- | --- | --- | --- | --- | --- | --- | --- | --- | --- | --- | --- | --- |
> > > > | Adaptation steps = 1 | 0.180 | 0.175 | 0.192 | 0.197 | 0.179 | 0.181 | 0.155 | 0.145 | 0.122 | 0.082 | 0.061 | 0.116 | 0.203 |
> > > > | Adaptation steps = 3 | 0.181 | 0.181 | 0.173 | 0.201 | 0.160 | 0.130 | 0.108 | 0.083 | 0.075 | 0.062 | 0.047 | 0.098 | 0.391 |
> > > > | Adaptation steps = 10 | 0.186 | 0.175 | 0.178 | 0.179 | 0.140 | 0.091 | 0.079 | 0.068 | 0.057 | 0.050 | 0.038 | 0.098 | 0.946 |
> > > >
> > > > | MR-MAML Test Loss, lr$_b$ = | 0.001 | 0.002 | 0.005 | 0.01 | 0.02 | 0.05 | 0.1 | 0.2 | 0.5 | 1 | 2 | 5 | 10 |
> > > > | --- | --- | --- | --- | --- | --- | --- | --- | --- | --- | --- | --- | --- | --- |
> > > > | Adaptation steps = 1 | 0.174 | 0.174 | 0.173 | 0.172 | 0.170 | 0.170 | 0.160 | 0.156 | 0.138 | 0.128 | 0.126 | 0.143 | 0.158 |
> > > > | Adaptation steps = 3 | 0.168 | 0.169 | 0.170 | 0.171 | 0.165 | 0.157 | 0.148 | 0.133 | 0.125 | 0.125 | 0.119 | 0.130 | 0.233 |
> > > > | Adaptation steps = 10 | 0.171 | 0.174 | 0.168 | 0.169 | 0.158 | 0.141 | 0.126 | 0.124 | 0.123 | 0.115 | 0.116 | 0.124 | 0.132 |
> > > >
> > > > | PAC-BUS Test Loss, lr$_b$ = | 0.001 | 0.002 | 0.005 | 0.01 | 0.02 | 0.05 | 0.1 | 0.2 | 0.5 | 1 | 2 | 5 | 10 |
> > > > | --- | --- | --- | --- | --- | --- | --- | --- | --- | --- | --- | --- | --- | --- |
> > > > | Adaptation steps = 1 | 0.170 | 0.176 | 0.174 | 0.166 | 0.173 | 0.166 | 0.165 | 0.150 | 0.138 | 0.127 | 0.126 | 0.142 | 0.146 |
> > > > | Adaptation steps = 3 | 0.170 | 0.173 | 0.174 | 0.172 | 0.167 | 0.160 | 0.150 | 0.134 | 0.126 | 0.120 | 0.121 | 0.126 | 0.148 |
> > > > | Adaptation steps = 10 | 0.178 | 0.170 | 0.170 | 0.170 | 0.157 | 0.142 | 0.126 | 0.123 | 0.125 | 0.117 | 0.116 | 0.123 | 0.128 |
> > > >
> > > > Next, we present the bounds for MLAP, MR-MAML, and PAC-BUS for the same set of hyper-parameters.
> > > >
> > > > | MLAP-M Bound, lr$_b$ = | 0.001 | 0.002 | 0.005 | 0.01 | 0.02 | 0.05 | 0.1 | 0.2 | 0.5 | 1 | 2 | 5 | 10 |
> > > > | --- | --- | --- | --- | --- | --- | --- | --- | --- | --- | --- | --- | --- | --- |
> > > > | Adaptation steps = 1 | **1.002** | 1.002 | 1.002 | 1.003 | 1.006 | 1.051 | 1.281 | 1.539 | 2.418 | 3.252 | 4.21 | 10.023 | 21.793 |
> > > > | Adaptation steps = 3 | 1.002 | 1.002 | 1.003 | 1.009 | 1.024 | 1.251 | 1.767 | 2.756 | 3.789 | 3.927 | 5.603 | 11.131 | 27.283 |
> > > > | Adaptation steps = 10 | 1.003 | 1.004 | 1.016 | 1.050 | 1.175 | 1.795 | 2.441 | 3.306 | 4.943 | 6.065 | 7.687 | 17.683 | 47.946 |
> > > >
> > > > | MR-MAML Bound, lr$_b$ = | 0.001 | 0.002 | 0.005 | 0.01 | 0.02 | 0.05 | 0.1 | 0.2 | 0.5 | 1 | 2 | 5 | 10 |
> > > > | --- | --- | --- | --- | --- | --- | --- | --- | --- | --- | --- | --- | --- | --- |
> > > > | Adaptation steps = 1 | 0.347 | 0.347 | 0.346 | 0.348 | 0.342 | 0.342 | 0.334 | 0.329 | 0.310 | 0.300 | 0.298 | 0.319 | 0.332 |
> > > > | Adaptation steps = 3 | 0.343 | 0.348 | 0.344 | 0.347 | 0.341 | 0.332 | 0.320 | 0.308 | 0.299 | 0.297 | 0.293 | 0.302 | 0.394 |
> > > > | Adaptation steps = 10 | 0.345 | 0.346 | 0.342 | 0.342 | 0.333 | 0.314 | 0.304 | 0.300 | 0.297 | 0.293 | **0.288** | 0.299 | 0.306 |
> > > >
> > > > | PAC-BUS Bound, lr$_b$ = | 0.001 | 0.002 | 0.005 | 0.01 | 0.02 | 0.05 | 0.1 | 0.2 | 0.5 | 1 | 2 | 5 | 10 |
> > > > | --- | --- | --- | --- | --- | --- | --- | --- | --- | --- | --- | --- | --- | --- |
> > > > | Adaptation steps = 1 | 0.200 | 0.208 | 0.214 | 0.211 | 0.217 | 0.213 | 0.206 | 0.197 | 0.180 | 0.168 | **0.168** | 0.185 | 0.192 |
> > > > | Adaptation steps = 3 | 0.214 | 0.230 | 0.247 | 0.254 | 0.250 | 0.242 | 0.231 | 0.215 | 0.207 | 0.204 | 0.201 | 0.206 | 0.229 |
> > > > | Adaptation steps = 10 | 0.309 | 0.350 | 0.378 | 0.387 | 0.378 | 0.362 | 0.350 | 0.346 | 0.342 | 0.338 | 0.335 | 0.341 | 0.349 |
> > > >
> > > > These results show the dependence of the PAC-BUS upper bound (specifically the uniform stability regularizer term $\beta_{US}$) has on the learning rate and number of base-learning update steps whereas the bound for MR-MAML does not suffer with increasing base-learning steps or learning rate. We note, however, that the qualitative results in the paper remain unchanged: the tightest guarantee obtained using PAC-BUS is significantly stronger than those for any tuning of MR-MAML and MLAP-M. We bold the tightest guarantee achieved in the tables above to highlight this. Thank you for the suggestion to make this comparison.
> > > >
> > > > **Bounds of Table 2:**
> > > > The bound we use for few shot learning is indeed $\mathcal{O}(\frac{1}{\sqrt{l}} + \frac{m}{m+n})$. However, $n$ here refers to the number of validation samples, not the number of adaptation samples (samples which are used to update the learner at the base-level). $m$ is the number of adaptation samples. Thus for fixed $n$, $\frac{m}{m+n}$ increases as $m$ increases. Also note that in the settings we consider, $n$ is much larger than $m$, so the increase can be a significant proportion. However, as you point out, the parameters for the gradient descent also affect this term and the tuning was different for each of $m = \{1, 3, 5\}$ (we note the hyperparameters in Appendix A.8.2). We believe this was also a factor in the bound increasing with $m$.
> > > >
> > > > We present Theorem 3 in the classic multi-task learning setting and do believe the framework is applicable to that setting. However, we see the ability of our bound to take advantage of validation data and reduce the size of the uniform stability regularizer as an advantage for few-shot learning.
> > > >
> > > >
> > > > **Size of the validation set:**  The few-shot learning setting we consider (where the number of adaptation samples is increased, not the number of tasks) is consistent with both [27] and [58]. Since we also compare to methods presented in these papers, we felt that this was an appropriate few-shot learning setting. Note that we do not assume access to validation data at test time, i.e. the base-learner adapts with a few samples at test time and the guarantee holds (in expectation) for new samples which come from that task.
> > > >
> > > >
> > > > **Comparisons to other methods:**  We believe that allowing for validation data when training for the few-shot learning case is important for providing tight generalization bounds. Indeed we show that MR-MAML (which can incorporate validation data) provides a significantly tighter guarantee as compared with MLAP (which cannot) while otherwise remaining relatively similar. It is not clear from [46] that this is possible, but an interesting direction for future work.  Additionally, as we mention in the paper, it is challenging to compute a tight sample convergence bound for MLAP. The same challenge would arise with the method presented in [46]. However, we agree that their additional divergence measure is a good method of regularization and  was shown to provide strong results in terms of empirical test loss. Since we aimed to focus on proving tighter generalization guarantees (and not necessarily improving empirical test performance) in this work, we compare to a broad range of generalization bounds. We believe that the MLAP method is a good baseline for PAC-Bayes generalization guarantees for meta-learning, the FLI-Batch [27] bound is a strong comparison for a bound which is not based on PAC-Bayes, and MR-MAML provides a recent and already-popular PAC-Bayes bound for meta-learning.
> > > >
> > > >
> > > > **Editing comment:**  We agree and have presented a modified notion of stability as part of the first reply (see above). We also present a version of Theorem 2 which helps with the exposition in the proof. We will include these in the revision as well as an explanation of how they relate to our setting and are different from previous definitions.

---

> > > > ### Comment · Reviewer_wSzt · 2021-09-02
> > > > **Many thanks but I think that the proof of Eq(1) is false or at least not correctly justified**
> > > >
> > > > Dear authors,
> > > >
> > > > I really thank you for providing this long and detailed reply.
> > > >
> > > > As a quick answer, I just focus on the concern I still have: unless I am mistaken or I missed someething: for me the result of Eq(1) is false or at least wrongly justified.
> > > >
> > > > For me, from the McAllester Theorem, you could get the following result (sorry if the notations are not correct):
> > > > $$E_{t \sim T}  E_{S\sim Z_{t}^{m}} E_{\theta\sim \Theta} f_{A}(\theta,S)\leq \frac{1}{l} \sum_{i=1}^{l} E_{S\sim Z_{i}^{m}} E_{\theta\sim\Theta}  f_{A}(\theta,S) + \sqrt{\frac{D_{KL}(\Theta\mid  \Theta_{0})+\ln{\frac{2\sqrt{l}}{\delta}}}{2l}}$$
> > > >
> > > > But you cannot have directly the empirical error with respect to the samples of tasks in the result, ie $\frac{1}{l}\sum_{i=1}^{l} E_{\theta\sim\Theta} f_{A}(\theta,S_i)$.
> > > > This would imply to consider the $m$ additional variables associated to the $m$ examples for each task and then the examples would not be identically distributed because of the different task distributions involved, thus you cannot use the tools from Maurer or McAllester (at least directly).
> > > >
> > > > Maybe you can prove in another way and I would be happy to know, but the proof is at least incorrect. Unless strong assumptions, I'm not sure that the result can be obtained without having a looser bound, to be discussed (?)
> > > > By the way, this may explain why other Pac-Bayes results had included separate PAC-Bayes bounds for each task.
> > > >
> > > > My evaluation cannot change from the answer provided, unless the authors could justify their claim.
> > > > If I'm wrong or I missed something, do not hesitate.

---

> > > > > ### Author Response · Authors · 2021-09-03
> > > > > **Clarification of Proof Step 1**
> > > > >
> > > > > Thank you for your quick reply. We are grateful for your thorough evaluation of our responses and the opportunity to provide further clarifications.
> > > > >
> > > > > We believe Equation (1) in our previous response is correct and will provide further justification for it here. First, consider an arbitrary distribution $D$ over datasets $S$ of size $m$ (for now assume nothing about $D$; we will make this distribution concrete later). We then have the following bound:
> > > > >
> > > > > For data-independent prior distribution $\Theta_0$, loss function $f_A$ which satisfies Assumption 1, $l \geq 8$, and $\delta \in (0,1)$, with probability at least $1-\delta$ over a sampling of $\textbf{S} = \\{S_1, S_2, \dots, S_l\\} \sim D^l$, the following holds for all $\Theta$:
> > > > > \begin{equation}
> > > > > E_{S\sim D} \ E_{\theta \sim \Theta}\ f_A(\theta, S) \leq \frac{1}{l} \sum_{i=1}^l \ E_{\theta \sim \Theta} \ f_A(\theta, S_i) + \sqrt{\frac{D_{KL}(\Theta \| \Theta_0) + \ln\frac{2\sqrt{l}}{\delta}}{2l}}. \ \ \ \ \ \ \ \ \ \ (1)
> > > > > \end{equation}
> > > > >
> > > > > Datasets $S$ indeed must be sampled i.i.d from $D$ in order for the bound to hold; however, the bound above holds for any choice of $D$. In order to derive Equation (1) in our previous response, we choose $D$ to be the *marginal distribution* $D_S$ over datasets. This is the distribution over datasets one obtains by first sampling a task $t$ from $T$, and then sampling a dataset $S$ from $Z^m_t$. Next, consider the following equations (for simplicity, we use summations instead of integrals to compute expectations; $p(t)$ represents the probability of sampling $t$ and $p(S|t)$ is the probability of sampling $S$ given $t$):
> > > > >
> > > > > \begin{align}
> > > > >     E_{t\sim T} E_{S \sim Z^m_t} E_{\theta \sim \Theta} f_A(\theta, S) & = \sum_t p(t) \sum_S p(S|t) E_{\theta \sim \Theta} f_A(\theta, S) \\\\
> > > > >     & = \sum_{t,S} p(t) p(S|t) E_{\theta \sim \Theta} f_A(\theta, S) \\\\
> > > > >     & = \sum_{t,S} p(S, t) E_{\theta \sim \Theta} f_A(\theta, S) \\\\
> > > > >     & = \sum_{S} \bigg( E_{\theta \sim \Theta} f_A(\theta, S) \underbrace{\sum_{t} p(S, t)}_{= p(S)} \bigg).
> > > > > \end{align}
> > > > >
> > > > > Here $p(S)$ corresponds to the marginal distribution over datasets $S$. Note that the last line above holds because $E_{\theta \sim \Theta} f_A(\theta, S)$ doesn't depend on $t$. Thus we have
> > > > > \begin{align}
> > > > >     E_{t\sim T} E_{S \sim Z^m_t} E_{\theta \sim \Theta} f_A(\theta, S) & = \sum_S p(S) E_{\theta \sim \Theta} f_A(\theta, S) = E_{S \sim D_S} E_{\theta \sim \Theta} f_A(\theta, S).  \ \ \ \ \ \ \ \ \ \ (2)
> > > > > \end{align}
> > > > > Note that the RHS of (2) is identical in form to the LHS of the generic PAC-Bayes bound in (1). We thus obtain Equation (1) in our previous response by choosing $D$ to be the marginal distribution $D_S$. We are happy to clarify the explanation of the marginal distribution in the revision and include the explicit steps above in the appendix.

---

> > > > > > ### Comment · Reviewer_wSzt · 2021-09-03
> > > > > > **Thank you very much  - I'm convinced  now - sorry for the misunderstanding**
> > > > > >
> > > > > > Dear authors,
> > > > > >
> > > > > > Thank you for your precisions. I actually did not completely understand your setting and this now clear how you deal with what you called the marginal distribution. I'm convinced that the results are correct and the way you deal with the tasks is indeed interesting.
> > > > > >
> > > > > > Thank you for the additional results which complete the work in positive manner. I think that studying more the algorithm parameters such as the number of iterations or specific parameters is interesting in your context.
> > > > > >
> > > > > > Having a lot of validation data seems not always obvious in a few-shot learning setting, but when this is possible it brings some interesting information. In addition with the potential increasing of the bounds, these are the last points from which I have some reservations.
> > > > > >
> > > > > > I hope that you will update your paper accordingly which will help to better clarify your contribution.
> > > > > >
> > > > > > I increase then my score.

---

> ### Author Response · Authors · 2021-09-03
> **Thank you for the valuable discussion**
>
> We thank the reviewer for the opportunity to have a thorough and valuable discussion. We are very grateful for the time and care that the reviewer has taken in this exchange.
>
> We are happy that the reviewer appreciates the quality of the work and the additional studies and clarifications we have provided in the discussion period. We will add the results and clarifications from the discussion to the revised version of the paper.

---

### Official Review · Reviewer_Q1zC · 2021-07-16

**Rating:** 6
**Confidence:** 3

**Summary:**

The paper proposes a generalization bound for meta-learning based on the following PAC frameworks: Uniform Stability for assessing the generalization of the base (task-specific) models to unseen data, and PAC-Bayes for estimating the generalization of the meta-learner to unseen tasks.
This implies the use of a stochastic meta-learner, that provides the distribution from which base models are sampled for initializing the task learner.
The obtained bound is then tightened for the few-shot learning context, by accounting for additional validation data.
The derived bound is finally used as objective function for optimizing the posterior model distribution by stochastic gradient descent, for the particular case of Gaussian posterior and prior distributions and using the reparameterization trick.

**Limitations And Societal Impact:**

Yes

**Main Review:**

The paper's contributions build on works on bi-level generalization guarantees. To the best of my knowledge, the combination of uniform stability and PAC-Bayes frameworks is new in the meta-learning literature.

The paper is generally well written, but two key points should be clarified:

1. The notion of uniform stability is used to derive a generalization guarantee in expectation, contrary to the original result from Bousquet which is a PAC guarantee, hence standing with a certain probability.
This is a key aspect of the approach and should be highlighted, as it allows to derive the bound of Theorem 3 for meta-learning without using a union bound. Indeed, as highlighted in the paper relying on a union bound for deriving the generalization guarantees is a disadvantage of existing works, as it results in looser guarantees.
I would also suggest to explicitly report in the theorems the parameters on which \beta_{US} depends, in order to make it easier for the reader to assess the bound.

2. In Algorithm 1, the posterior's mean and variance are updated via a Monte Carlo estimate of the expected loss, by drawing a model per task to then update it with the base-learner algorithm and to be used to estimate the bound. However I believe that this procedure is unbiased only if the base-learner algorithm is differentiable and jointly optimized with the posterior. It is not clear if this is the case in the proposed optimization procedure.

The main downside of the derived bounds is that the convergence of the PAC-Bayesian complexity terms explicitly depends only on the number of tasks and not on the sizes of the task samples. This is a source of vacuity on real problems, as the number of tasks is usually small.
Indeed, in the experiments with the Omniglot dataset of the paper, the complexity term has to be reduced by using smaller models and by down-scaling it.
A simple (but limited) improvement could be achieved by using PAC-Bayesian bounds with faster convergence rate, such as Seeger's bound [Seeger][Maurer].

Finally, in the experimental evaluation, it would be useful to report also the test error of a model trained on each task singularly (averaged over all the tasks). This would be a valuable sanity check, to make sure that the proposed meta-learning algorithm does not degrade the baseline performance.

### Minor:
In Table 2 PAC-BUS bound increases when the number of points m increases. Shouldn't it be the reverse?

[Seeger] PAC-Bayesian Generalisation Error Bounds for Gaussian Process Classification, JMLR, 2002.

[Maurer] A note on the PAC Bayesian theorem, 2004

### UPDATE
The authors addressed well my questions, but after reading also the other reviews I am still not sure whether recommending acceptance or rejection of the paper, and I am keeping my initial rating of 6. The work improves upon state-of-the-art generalization guarantees for meta-learning and in particular in the few-shot learning context and the paper is well written. However, I still have the following concerns:

1. the novelty of the contributions is limited, as the proposed bound is a modification of works by [Pentina et al.] using (known) stability results at the base-level instead of a PAC-Bayes guarantee;
2. the practical impact of the bound itself is also limited as at the meta-level it depends only on the number of tasks; this is an intrinsic problem of bi-level guarantees, and the solutions proposed by the authors can only marginally alleviate it.

**Time Spent Reviewing:**

8

---

> ### Author Response · Authors · 2021-08-10
> **Response to Reviewer Q1zC**
>
> Thank you very much for your thoughtful feedback and valuable suggestions. We address your comments and suggestions below.
>
> 1. We agree that using uniform stability for a generalization guarantee in expectation is a key aspect of our approach for deriving the meta-learning generalization bound. We will drive home this point further in the revised version of the paper. Reporting the parameters on which $\beta_{US}$ depends in the theorems is also a good idea — thank you for the suggestion.
>
> 2. Thank you for mentioning this. We agree that this estimator needs justification as unbiased, so we will clarify here and expand upon this point in the revision. For fixed and deterministic algorithm A, dataset S, and sample z, consider the bias of the proposed estimator:
> $$ E_{\xi \sim \mathcal{N}(0, I)} L(h_{A(\mu + \sqrt{s}\odot \xi,S)}, z) - E_{\theta \sim \Theta} L(h_{A(\theta, S)}, z).$$
> Since we let $\Theta$ be the multivariate Gaussian distribution $\mathcal{N}(\mu, s)$, sampling $\theta \sim \mathcal{N}(\mu, s)$ is equivalent to $\theta \leftarrow \mu + \sqrt{s}\odot\xi, \ \xi \sim \mathcal{N}(0, I)$. Thus, we may write
> $$ E_{\xi \sim \mathcal{N}(0, I)} L(h_{A(\mu + \sqrt{s}\odot \xi,S)}, z) - E_{\theta \leftarrow \mu + \sqrt{s}\odot\xi, \xi \sim \mathcal{N}(0, I)} L(h_{A(\theta, S)}, z) = 0.$$
> Since this holds for any sample $z$, we can extend to use the loss on a dataset. From this point, there is correspondence to the optimization method presented in [17], on which we base our procedure. Thus, we believe this does satisfy the requirements for an unbiased estimator.
>
>
> We agree that the number of tasks being small can lead to the PAC-Bayes complexity terms being large and make them looser. We are in fact already using Maurer’s PAC-Bayes bound in our implementation and will clarify this point in the revision. There are also more recent PAC-Bayes bounds [45, 53, mentioned briefly on L.193-194], which could afford further (modest) improvements. Another potential direction to mitigate the effects of having a small number of tasks is to leverage prior knowledge on the distribution of tasks and embed this prior knowledge through the PAC-Bayes prior; indeed, the ability to specify a prior is one of the appealing features of the PAC-Bayes bound. We will address these points in the revision.
>
>
> Thank you for the suggestion to report the test errors of models trained on each task singularly. This is a good sanity check and we will add this in the revision.
>
>
> Table 2: We minimize the bound presented in Inequality (9) to produce the PAC-BUS bound results in Table 2. Note that the 3rd term of the upper bound increases with increasing $m$ when the number of validation samples $n$ is kept consistent. Thus, the regularizer penalizes adaptation with more samples in the base-learning step. The penalty was large enough to offset improvements in the training loss. The result is that the PAC-BUS bound is larger when $m$ increases for this specific example.

---

### Official Review · Reviewer_dWUk · 2021-07-16

**Rating:** 7
**Confidence:** 3

**Summary:**

This work introduces a  novel PAC-Bayes bound for gradient-based meta-learning algorithms. This PAC-Bayes bound exploits a previous bound for uniformly stable learning algorithms.  In opposite to previously proposed bounds for meta-learning methods, the bound is non-vacuous in some data sets and much tighter. Authors also show that a heuristic method based on the minimization of the proposed bound leads to novel meta-learning algorithms.

**Ethical Concerns:**

No ethical concerns.

**Limitations And Societal Impact:**

The limitations of the paper are discussed as well as the societal impact.

**Main Review:**

The theoretical study of meta-learning algorithms is a very relevant topic in machine learning due to the wide range of potential applications. This work proposes a novel PAC-Bayes bounds that builds in two different frameworks to compute performance guarantees: PAC-Bayes theory and uniformly stable algorithms. Although the uniformly stability bound was previously proposed, this connection seems novel to me.

The paper is very well written and easy to follow despite the technical difficulty of the topic. And the proposed bound is thoroughly evaluated and gives rise to a novel class of bounds that are non-vacuous in some relevant cases. The derivation and evaluation of a novel meta-learning algorithm based on the minimization of the proposed PAC-Bayes bound are also worth noticing.

**Time Spent Reviewing:**

3

---

> ### Author Response · Authors · 2021-08-10
> **Response to Reviewer dWUk**
>
> Thank you very much for your valuable feedback. We appreciate your positive comments on our work.

---

> > ### Comment · Reviewer_dWUk · 2021-09-04
> > **Post Rebuttal Comment**
> >
> > After reading the other reviews, the authors rebuttal and the discussions, I keep my score and I support the acceptance of this paper.

---

### Decision · Program_Chairs · 2021-09-27

**Decision:**

Accept (Poster)

**Comment:**

The reviewers and myself all agree on the significance of the results and their interest to the NeurIPS community. The reviews expressed a number of technical concerns which were all addressed satisfactorily by the authors during the discussion phase. If not done already, I would invite the authors to consider revising their manuscript along the many suggestions made by the reviewers.